# Learning Credal Ensembles via Distributionally Robust Optimization

**Kaizheng Wang** [1,2]   **Ghifari Adam Faza** [3,4]   **Fabio Cuzzolin** [5]   **Siu Lun Chau** [2]   **David Moens** [3,4]   **Hans Hallez** [1]

## Abstract

Credal predictors are epistemic-uncertainty-aware models that produce a convex set of probabilistic predictions. They provide a principled framework for quantifying predictive epistemic uncertainty (EU) and have been shown to improve model robustness across a range of settings. However, most state-of-the-art (SOTA) methods primarily define EU as disagreement induced by random training initializations, which mainly reflects sensitivity to optimization randomness rather than uncertainty from more substantive sources. In response, we formulate EU as disagreement between models trained under different degrees of relaxation of the i.i.d. assumption between the training and test distributions. Building on this idea, we propose *CreDRO*, which learns an ensemble of plausible models via distributionally robust optimization. As a result, CreDRO captures EU arising not only from training randomness but also from informative disagreement due to potential train–test distribution shifts. Empirically, CreDRO consistently outperforms SOTA credal approaches on downstream tasks, including out-of-distribution detection on extensive benchmarks and selective classification in medical settings.

## 1. Introduction

Quantifying predictive uncertainty in deep neural networks (NNs) has become increasingly important, as reliable uncertainty quantification (UQ) is essential for improving the robustness and trustworthiness of machine learning sys-

tems, particularly in safety-critical applications (Zhou et al., 2012; Mehrtens et al., 2023; Wang et al., 2025a). However, informative UQ requires distinguishing aleatoric uncertainty (AU), which stems from inherent randomness in the data-generation process, from epistemic uncertainty (EU), which arises from NNs' limited knowledge about the true input-output relationship (Hüllermeier & Waegeman, 2021). These two forms of uncertainty have fundamentally different implications for downstream decision-making. In particular, reliable estimation of EU is beneficial for tasks where utilizing model's predictive confidence is crucial, such as in selective prediction (Shaker & Hüllermeier, 2021b; Chau et al., 2025a), learning to reject and defer (Liu et al., 2022), Bayesian optimization (Tuo & Wang, 2022), and out-of-distribution (OOD) detection (Mucsányi et al., 2024).

Unlike AU, which is usually modeled by a single (conditional) probability distribution (e.g., via a softmax probabilistic prediction in classification), EU generally requires a second-order formalism to represent uncertainty about the model's (probabilistic) prediction itself (Hüllermeier & Waegeman, 2021; Wang et al., 2025a). Under this context, credal sets (Levi, 1980), i.e., convex sets of probability distributions, serve as an appealing second-order representation of uncertainty over first-order probabilistic predictions (Zaffalon, 2002; Corani & Zaffalon, 2008; Corani et al., 2012; Mauá et al., 2017), and have inspired recent advances to improve EU quantification in deep learning. For instance, a recent credal wrapper approach (Wang et al., 2025b) formulates credal set predictions from multiple softmax outputs of a deep ensemble (Lakshminarayanan et al., 2017) trained with different random initializations. In addition, a credal ensembling method (Nguyen et al., 2025) is proposed as an alternative post hoc extension to a classical ensemble. This method also allows discarding outlier probabilities through a hyperparameter $\alpha$, preventing credal sets from becoming too large. Moreover, Löhr et al. (2025) introduce a scheme that increases ensemble diversity and selects plausible members using a predefined threshold based on the concept of relative likelihood. See Appendix A for an extended literature review. Although these credal predictors have been shown to improve model robustness across a range of classification settings, they capture EU as disagreement induced by random training initializations, mainly reflecting the sensitivity to optimization randomness rather than uncertainty

This work was mainly conducted at KU Leuven and completed at Nanyang Technological University. [1]Department of Computer Science, KU Leuven, Belgium [2]College of Computing and Data Science, Nanyang Technological University, Singapore [3]Department of Mechanical Engineering, KU Leuven, Belgium [4]Flanders Make@KU Leuven, Belgium [5]School of Engineering, Computing and Mathematics, Oxford Brookes University, U.K.. Correspondence to: Kaizheng Wang <kaizheng.wang@ntu.edu.sg>.

*Proceedings of the 43$^{rd}$ International Conference on Machine Learning*, Seoul, South Korea. PMLR 306, 2026. Copyright 2026 by the author(s).

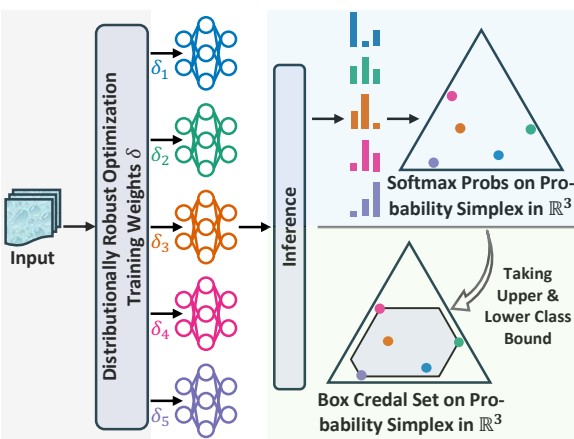

*Figure 1.* CreDRO Concept. ① Training: An ensemble is trained using distributionally robust optimization with members weighted differently to simulate varying degrees of train-test distribution shifts (see Section 3.1). ② Inference: Softmax probabilities are converted into class-wise probability intervals, creating a box credal set (see Section 3.2).

from more substantive sources.

**Contributions.** In response, we formulate EU as disagreement among ensemble member models trained under different degrees of relaxation of the i.i.d. assumption between the training and test distributions. Following this principle, we propose *CreDRO*, which learns an ensemble of plausible models using distributionally robust optimization (DRO) by varying a hyperparameter to simulate different degrees of potential train–test distribution shift. As a result, CreDRO captures not only training randomness but also informative disagreement arising from potential train–test distribution shifts. At inference time, CreDRO transforms individual softmax probabilities from the ensemble into class-wise probability intervals (De Campos et al., 1994), and uses them to form a box credal set (Wang et al., 2025b) prediction, i.e., a set of probabilities confined by the intervals. The concept of CreDRO is illustrated in Figure 1.

Compared to state-of-the-art (SOTA) credal classifiers and the deep ensemble baseline, CreDRO consistently achieves superior epistemic uncertainty (EU) quantification performance across multiple out-of-distribution (OOD) detection and selective classification benchmarks. This indicates that distributionally robust optimization (DRO) yields substantially higher-quality epistemic uncertainty estimates than those obtained from random model initializations.

**Further Related Work.** In addition to ensemble and credal predictors, Bayesian neural networks (BNNs) (Blundell et al., 2015; Gal & Ghahramani, 2016; Krueger et al., 2017; Mobiny et al., 2021) and evidential deep learning (EDL) (Malinin & Gales, 2018; 2019; Charpentier et al., 2020) are alternative second-order approaches. However, BNNs often face practical limitations, including scalability

issues with large datasets or complex model architectures and the need for many forward passes (Mukhoti et al., 2023). EDL has also attracted recent criticism (Bengs et al., 2023; Juergens et al., 2024; Shen et al., 2024). In particular, EDL may fail to represent EU faithfully. For example, Juergens et al. (2024) show that confidence bounds derived from EDL's EU estimates differ substantially from those of the reference distribution, which is defined as the distribution of the first-order predictor induced by the training data sampling process and is used to assess EU quantification quality. Singh et al. (2024) have proposed imprecise learning to consider all possible DRO constraints by solving an infinite-objective optimization, instead of retaining the uncertainty through an ensembling approach.

**Paper Outline.** The rest of this paper is organized as follows. Section 2 provides the background of DRO and deep ensembles. Section 3 introduces our CreDRO in full detail. Section 4 describes experimental validation. Section 5 summarizes the conclusion and future work.

**Conflict of Interest Disclosure.** The authors declare that they have no known competing financial interests or personal relationships that could have appeared to influence the work reported in this paper.

## 2. Preliminaries

### 2.1. Distributionally Robust Optimization (DRO)

In supervised learning, a neural network (NN) with trainable parameters, denoted as $h_\theta(\cdot)$, is generally trained on a set of labeled samples $\{\boldsymbol{x}_n, y_n\}_{n=1}^N$. One standard approach is the empirical risk minimization (ERM) framework, where the parameter $\theta$ is obtained by solving

$$\underset{\theta}{\text{minimize}} \left\{ \frac{1}{N} \sum_{n=1}^N \mathcal{L}\big((h_\theta(\boldsymbol{x}_n), y_n)\big) \right\}, \quad (1)$$

where $\mathcal{L}(\cdot, \cdot)$ is a chosen loss function. However, the ERM often yields over-optimistic predictions during deployment because it assumes the training and test distributions are identical—an assumption that frequently fails in practice, where test data can differ significantly from the training samples (Huang et al., 2022).

To improve the robustness of NNs to distribution discrepancies, the DRO framework relaxes the i.i.d. assumption between the training and test distributions by assuming that the future test distribution lies within a neighborhood of the training distribution $P$. Under this context, the DRO framework minimizes a worst-case expected risk $R(\theta)$ over an uncertain set of distributions $\mathcal{U}$ (Ben-Tal et al., 2013; Oren et al., 2019; Duchi et al., 2021).

$$\underset{\theta}{\text{minimize}} \left\{ R(\theta) \doteq \sup_{P \in \mathcal{U}} \mathbb{E}_{(\boldsymbol{x},y) \sim P} \mathcal{L}\big(h_\theta(\boldsymbol{x}), y\big) \right\}. \quad (2)$$

To make the DRO framework practical, *group DRO* methods have been proposed (Oren et al., 2019; Sagawa et al., 2019). These approaches assume the training distribution $P$ is a mixture of $m$ groups $P_g$, indexed by $g \in \mathcal{G} = \{1, ..., m\}$. Since the optimum of a linear program occurs at a vertex, the worst-case risk $R(\theta)$ in (2) simplifies to maximizing the expected loss over each group, with the training objective:

$$\underset{\theta}{\text{minimize}} \left\{ \underset{g \in \mathcal{G}}{\text{maximize}} \, \mathbb{E}_{(\boldsymbol{x}, y) \sim P_g} \mathcal{L}\big(h_\theta(\boldsymbol{x}), y\big) \right\}. \quad (3)$$

A remaining practical challenge in implementing (3) lies in constructing an empirical estimate of the distribution $P_g$ for each group when only limited training data are available.

In response, *adversarially reweighted learning*—a specific form of *group DRO* (Sagawa et al., 2019; Lahoti et al., 2020; Nam et al., 2020)—has been adopted. Specifically, (3) is implemented as a minimax game between a learner and an adversary. The learner optimizes parameters $\theta$ to minimize the expected loss whereas the adversary maximizes the expected loss by adversarially assigning the sample with weights $w_n$, collected in a vector $\boldsymbol{w}$. The resulting training objective becomes:

$$\underset{\theta}{\text{minimize}} \left\{ \underset{\boldsymbol{w} \in \mathbb{W}}{\text{maximize}} \, \frac{1}{N} \sum_{n=1}^{N} w_n \mathcal{L}\big(h_\theta(\boldsymbol{x}_n), y_n\big) \right\}, \quad (4)$$

The predefined set of weight vectors $\mathbb{W}$ is a subjective design choice due to the lack of knowledge about the actual test distribution, and has been defined in various ways, see Lahoti et al. (2020); Nam et al. (2020); Sagawa et al. (2019); Wang et al. (2024).

### 2.2. Deep Ensembles

Deep ensembles (DE) have demonstrated significant advantages in quantifying prediction uncertainty by independently training multiple randomly-initialized neural networks (Lakshminarayanan et al., 2017). Specifically, a DE consists of a set of classical neural networks $\{h_{\theta_i}(\cdot)\}_{i=1}^{M}$ with probabilistic predictions. This is the default setup used throughout the paper unless otherwise noted. For classification, each network produces individual softmax probabilities $\{\boldsymbol{p}_i\}_{i=1}^{M}$ at inference time.

To make a final precise prediction, a DE takes the averaged probability vector $\tilde{\boldsymbol{p}} = M^{-1} \sum_{i=1}^{M} \boldsymbol{p}_i$. EU is quantified by using the well-known *approximate mutual information* (Hüllermeier & Waegeman, 2021), as follows:

$$\sum_{k=1}^{C} -\tilde{p}_k \log \tilde{p}_k - \frac{1}{M} \sum_{i=1}^{M} \sum_{k=1}^{C} -p_{i,k} \log p_{i,k}. \quad (5)$$

Here, $\tilde{p}_k$ denotes the $k$-th element of the averaged probability vector $\tilde{\boldsymbol{p}}$. The expression for epistemic uncertainty in (5) arises from the standard decomposition of total predictive uncertainty into aleatoric and epistemic components (Hüllermeier & Waegeman, 2021). In this setting,

EU encoded in the Bayes framework (Hüllermeier et al., 2022) reflects an ensemble's disagreement induced by random training initializations.

DEs have long served as a strong baseline for UQ in deep learning. Several variants—such as batch ensembles (Wen et al., 2020), masked ensembles (Durasov et al., 2021), and packed ensembles (Laurent et al., 2022)—have been proposed to provide uncertainty estimates comparable to those of standard DEs, while significantly improving computational efficiency. However, these variants differ from our primary focus, which is to prioritize further improvements in uncertainty quantification—a consideration that is particularly critical in safety-critical applications. More recently, the credal wrapper (Wang et al., 2025b) has demonstrated that mapping deep ensemble predictions into a credal framework can substantially improve the resulting epistemic uncertainty (EU) quantification. Nevertheless, the uncertainty captured by this post-hoc transformation remains fundamentally tied to the same source of variability, namely, ensemble disagreement arising from random training initializations.

## 3. Main Method: CreDRO

This section details our CreDRO framework. Section 3.1 introduces how CreDRO is trained by the DRO strategy, and Section 3.2 presents the credal prediction generation and its EU qualification.

### 3.1. Training Procedure

The CreDRO method constructs an ensemble of plausible probabilistic models, each trained under a different degree of potential train–test distribution shift. The resulting disagreement among these models reflects epistemic uncertainty about the prediction at deployment, in contrast to classical ensemble disagreement arising solely from different random model initializations.

Methodologically, we adopt the adversarially reweighted learning (ARL) framework in (4) from the *group DRO* family (see Section 2.1). Our method enables standard batchwise optimization for NN training under the ARL framework while encouraging each network to specialize in different group distributions. As a result, the CreDRO produces diverse probabilistic predictions that reflect a range of different degrees of relaxation of the i.i.d. assumption between train-test distributions. In the following, we first describe our implementation of the DRO strategy for a single model, and then present the ensemble training procedure.

**A Flexible ARL-based DRO Implementation.** We instantiate the uncertainty set $\mathbb{W}$ in (4) as a conditional value at risk (CVaR) set at level $\delta$ (Levy et al., 2020):

$$\mathbb{W} = \left\{ \mathbf{w} \geq 0 \mid \sum_{n=1}^{N} w_n = N, w_n \leq \delta^{-1}, \forall n \right\}. \quad (6)$$

A smaller $\delta$ corresponds to a strictly more conservative uncertainty set, i.e., $\delta_1 < \delta_2 \implies \mathbb{W}(\delta_1) \supseteq \mathbb{W}(\delta_2)$. By assigning each member a distinct $\delta_i$, CreDRO constructs models with formally distinct robustness profiles, rather than merely reweighting samples arbitrarily. Under this instantiation, the inner maximization in (4) admits a closed-form solution—the optimal adversary assigns weight $\delta^{-1}$ to the top-$\lfloor \delta N \rfloor$ loss samples and zero otherwise:

$$\underset{\mathbf{w} \in \mathbb{W}}{\text{maximize}} \frac{1}{N} \sum_{n=1}^{N} w_n \mathcal{L}_n\big(h_\theta(\boldsymbol{x}_n), y_n\big) \\ = \frac{1}{\delta N} \sum_{n \in S_\delta} \mathcal{L}_n\big(h_\theta(\boldsymbol{x}_n), y_n\big), \quad (7)$$

where $S_\delta$ denotes the top-$\delta$ highest-loss index set. This closed-form solution follows from solving a linear program over the simplex, where the optimum concentrates mass on the worst-$\delta$ fraction, and motivates the batch-wise top-$\delta$ approximation adopted in CreDRO.

Concretely, CreDRO adopts the approximation scheme of Huang et al. (2022); Wang et al. (2024): only the top-$\delta$ portion of samples with the highest loss in each batch are used for backpropagation, implicitly assigning $w_n > 1$ to selected samples and $w_n = 0$ to the rest. Batch-wise top-$\delta$ selection, therefore serves as a direct empirical approximation of the full-batch CVaR objective. Furthermore, as shown in prior work (Sagawa et al., 2019; Levy et al., 2020; Huang et al., 2022), these selected hard-to-learn instances may correspond to minority groups within the training data, thereby simulating potential domain shifts at test time.

**CreDRO Ensemble Training.** In the proposed scheme, a particular choice of $\delta$ represents a hypothesized level of discrepancy between the train-test distributions. To build an end-to-end CreDRO framework—i.e., an ensemble of plausible members that encode varying degrees of assumptions about possible distribution shifts—we introduce a global, user-defined hyperparameter $\delta_G \in [0.5, 1)$ that reflects the assumed worst-case divergence. The value of $\delta_i$ used to train the $i$-th individual model is then:

$$\delta_i = \frac{(1 - \delta_G)}{(M - 1)} \cdot (i - 1) + \delta_G, \quad (8)$$

which corresponds to a uniform interpolation over $[\delta_G, 1]$. Uniform interpolation is a natural, assumption-free design choice: in the absence of domain-specific knowledge favoring any particular $\delta$, all values are treated as equally plausible, ensuring ensemble members are evenly spread across the design range with no region over- or under-represented. We further validate this choice in Appendix D, where we compare uniform, exponential-like (concentrated near $\delta_G$), and logarithmic-like (concentrated near 1) interpolation strategies, demonstrating that CreDRO's performance is robust to this design choice.

---

**Algorithm 1** Batch-wise CreDRO Training Procedure
**Input:** Training batch data $\{\boldsymbol{x}_n, y_n\}_{n=1}^{\eta}$; hyperparameter $\delta_G \in [0.5, 1)$; batch size $\eta$; ensemble size $M$
**Output:** Trained $M$ single models $\{h_{\theta_i}\}_{i=1}^{M}$
**for** $i = 1, ..., M$ **do**
    **1.** Calculate loss $\mathcal{L}\big(h_{\theta_i}(\boldsymbol{x}_n), y_n\big)$ for each sample
    **2.** Sort the sample indices $(m_1, ..., m_\eta)$ in descending order of $\mathcal{L}\big(h_{\theta_i}(\boldsymbol{x}_n), y_n\big)$
    **3.** Compute $\delta_i$ from $\delta_G$ for the $i$-th model using (8)
    **4.** Define $\eta_{\delta_i} = \lfloor \delta_i \eta \rfloor$
    **5.** Minimize $\frac{1}{\eta_{\delta_i}} \sum_{j=1}^{\eta_{\delta_i}} \mathcal{L}\big(h_{\theta_i}(\boldsymbol{x}_{m_j}), y_{m_j}\big)$
**end for**

---

The lower bound of the design range for $\delta_G$ is set to 0.5, as an overly small $\delta_G$ would force the lowest-indexed ensemble member to train on an excessively small subset of high-loss samples (e.g., only 38 samples per batch of 128 when $\delta_G = 0.3$), leading to large and unstable gradient updates that can destabilize training, particularly in the early stages. When $\delta_G$ approaches 1, the worst-case assumes a less pronounced divergence between train-test distributions. If $\delta_G$ were to be set to 1, all samples would be selected for backpropagation, implying that $w_n = 1$ for any $n$ in (4). Consequently, the DRO loss components of all ensemble members would reduce to the ERM loss in (1), and CreDRO training would align with the standard ensemble.

In this work, $\delta_G$ is set to 0.5 by default to reflect a balanced degree of the train-test divergence and assess how this value helps our model outperform the baselines. In addition, our ablation study in Section 4.3 shows that CreDRO's performance is robust to the choice of hyperparameter $\delta_G$.

The batch-wise CreDRO training procedure is outlined in Algorithm 1. Here, $\mathcal{L}(\cdot, \cdot)$ could be any suitable loss, as long as it enables neural networks to produce probabilistic predictions, such as the cross-entropy (CE) loss or focal loss (Lin et al., 2017). In this work, we adopt the widely used CE loss to ensure fair and consistent comparison with existing baselines.

**Distinctions from the CreDE Baseline.** The method most closely related to our work is the recent credal deep ensemble (CreDE) approach (Wang et al., 2024), which also incorporates distributionally robust optimization (DRO) principles during training. However, several key distinctions set CreDRO apart. ① *Architecturally*, CreDE requires doubling the number of output neurons in the final layer in order to predict lower and upper probability bounds for each class. In contrast, CreDRO builds upon standard neural network architectures and requires *no architectural modifications*, thereby reducing model complexity and improving compatibility with existing training paradigms (see Table 3). ② Regarding individual ensemble members, CreDE uses

the DRO loss (4) to train the lower probability bound and the ERM loss (1) for the upper bound. In contrast, our approach applies DRO to train a classical NN, representing a relaxed form of the i.i.d. training assumption. Furthermore, since CreDE's upper and lower probability vectors are not normalized, it is restricted to one-hot label data when employing the CE loss (Wang et al., 2024), whereas CreDRO *does not suffer from this limitation*. ③ Regarding ensembling, CreDE uses a single fixed DRO hyperparameter, implicitly assuming the same train–test distribution divergence for all ensemble members. Consequently, disagreement among ensemble members arises mainly from random initialization. In contrast, CreDRO assigns *a range of* DRO hyperparameters to individual models, encouraging diverse probabilistic predictions that reflect varying relaxations of the i.i.d. assumption during training.

### 3.2. Credal Prediction and Uncertainty Quantification

**Generating Prediction.** For $C$-class classification problem, CreDRO transforms individual softmax probabilities $\{\boldsymbol{p}_i\}_{i=1}^M$ into class-wise probability intervals as follows:

$$\overline{p}_k = \max_{i=1,...,M} p_{i,k} \quad \underline{p}_k = \min_{i=1,...,M} p_{i,k}, \quad (9)$$

where $\overline{p}_k$ and $\underline{p}_k$ denote the upper and lower probability interval bounds for the $k$-th class, and $p_{i,k}$ is the $k$-th element of the given $\boldsymbol{p}_i$. These intervals induce a non-empty box credal set $\mathcal{K}_B$ as follows (De Campos et al., 1994):

$$\mathcal{K}_B = \left\{ \boldsymbol{p} \mid p_k \in [\underline{p}_k, \overline{p}_k] \,\forall k, \, \sum_{k=1}^C p_k = 1 \right\}. \quad (10)$$

$\mathcal{K}_B$ ensures a convex set of probability vectors, with each class probability value constrained to the specified interval.

There exists an alternative way to construct credal sets by taking the convex hull of a collection of probability vectors, denoted $\mathcal{K}_C$, as follows:

$$\mathcal{K}_C = \left\{ \sum_{i=1}^M \pi_i \boldsymbol{p}_i \mid \pi_i \geq 0, \sum_{i=1}^M \pi_i = 1 \right\}. \quad (11)$$

$\mathcal{K}_C$ is contained in the box credal set $\mathcal{K}_B$, i.e., $\mathcal{K}_C \subseteq \mathcal{K}_B$, since the convex hull is the smallest convex set containing the given collection of probability vectors. Nonetheless, for binary classification, $\mathcal{K}_C = \mathcal{K}_B$ as both are reduced to the same probability interval. From a computational perspective, however, $\mathcal{K}_B$ enables more efficient computation of EU, as discussed in Appendix G.1. Accordingly, we adopt the box credal set $\mathcal{K}_B$ in (10). Furthermore, the ablation study in Section 4.4 demonstrates that $\mathcal{K}_B$ consistently outperforms $\mathcal{K}_C$ across several OOD detection benchmarks.

**Uncertainty Quantification.** We quantify EU of our credal prediction $\mathcal{K}_B$ by computing the difference between upper and lower Shannon entropy (Abellán et al., 2006), written

as $\overline{H}(\mathcal{K}_B) - \underline{H}(\mathcal{K}_B)$. Computing $\overline{H}(\mathcal{K}_B)$ for CreDRO requires optimizing the following:

$$\begin{aligned} &\text{maximize} \sum_{k=1}^C -p_k \log p_k \\ &\text{s.t. } \sum_{k=1}^C p_k = 1 \text{ and } p_k \in [\underline{p}_k, \overline{p}_k] \text{ for } k = 1, ..., C \end{aligned}, \quad (12)$$

which searches for $\boldsymbol{p}$ within $\mathcal{K}_B$ to maximizes the entropy; computing $\underline{H}(\mathcal{K}_B)$ requires minimization instead. Both problems can be efficiently solved using the SciPy tool (Virtanen et al., 2020), and incur only marginal computational overhead (Wang et al., 2024; 2025b). Note that uncertainty quantification for credal sets remains an active research area; we consider alternative measures in the extended literature review (Appendix A).

## 4. Experimental Validation[1]

### 4.1. Comparison with SOTA Credal Classifiers and DE

Since groundtruth EU does not exist, *OOD detection* is commonly used as a practical benchmark for evaluate the quality of EU quantification, where stronger OOD detection performance indicates more informative uncertainty quantified (Wang et al., 2024; Löhr et al., 2025). In this setting, OOD detection is formulated as a binary classification problem, with in-distribution (ID) and OOD samples assigned to classes 0 and 1, respectively. The model's uncertainty estimate is then used as the prediction score, and the performance is evaluated via the area under the receiver operating characteristic curve (AUROC).

**Setup.** We benchmark our CreDRO against several SOTA credal classifiers considered in a recent study (Löhr et al., 2025): credal Bayesian deep learning (CreBNN) (Caprio et al., 2024), credal deep ensembles (CreDE) (Wang et al., 2024), the credal wrapper (CreWra) (Wang et al., 2025b), credal ensembling (CreEns) (Nguyen et al., 2025), and credal predictions based on relative likelihood (CreRL) (Löhr et al., 2025). We also include two strong ensemble baselines: a standard deep ensemble (DE), and *EN-DRO*—an ensemble *trained with our DRO framework* in Algorithm 1 but *without generating credal predictions*.

The experiment uses CIFAR10 (Krizhevsky et al., 2009) as the ID dataset, against several OOD datasets including SVHN (Hendrycks et al., 2021), Places365 (Zhou et al., 2018), CIFAR100 (Krizhevsky, 2012), FMNIST (Xiao et al., 2017), and ImageNet (Deng et al., 2009). For a fair comparison, we follow the setup of Löhr et al. (2025) and continue training an ensemble ($M = 20$, $\delta_G = 0.5$) of ResNet18 models (He et al., 2016) on the CIFAR10 training data using our CreDRO method. The performance of CreRL and

---

[1]Code implementation is available at `https://github.com/Kaizheng-WANG/Learning-Credal-Ensembles-via-Distributionally-Robust-Optimization`.

*Table 1.* AUROC score (%) for OOD detection based on EU using CIFAR10 as ID dataset. Best scores are bold. All results are averaged over 3 runs for fair comparison with existing baselines.

|  | SVHN | Places | CIFAR100 | FMNIST | ImageNet |
|---|---|---|---|---|---|
| DE | $94.8_{\pm 0.3}$ | $90.0_{\pm 0.2}$ | $90.6_{\pm 0.0}$ | $92.9_{\pm 0.3}$ | $88.9_{\pm 0.1}$ |
| EN-DRO | $95.7_{\pm 0.0}$ | $91.1_{\pm 0.1}$ | $91.6_{\pm 0.1}$ | $94.0_{\pm 0.1}$ | $90.0_{\pm 0.1}$ |
| **CreDRO** | $\mathbf{97.4}_{\pm 0.1}$ | $\mathbf{92.7}_{\pm 0.1}$ | $\mathbf{92.5}_{\pm 0.1}$ | $\mathbf{96.4}_{\pm 0.0}$ | $\mathbf{91.1}_{\pm 0.1}$ |
| CreWra | $95.7_{\pm 0.3}$ | $91.6_{\pm 0.1}$ | $91.6_{\pm 0.0}$ | $95.2_{\pm 0.0}$ | $89.0_{\pm 0.1}$ |
| CreRL$_{1.0}$ | $94.8_{\pm 0.3}$ | $91.8_{\pm 0.2}$ | $91.6_{\pm 0.1}$ | $95.7_{\pm 0.2}$ | $88.9_{\pm 0.2}$ |
| CreEns$_{0.0}$ | $95.5_{\pm 0.1}$ | $91.3_{\pm 0.0}$ | $91.4_{\pm 0.1}$ | $94.9_{\pm 0.1}$ | $88.8_{\pm 0.0}$ |
| CreDE | $94.3_{\pm 0.3}$ | $91.8_{\pm 0.0}$ | $91.2_{\pm 0.0}$ | $95.1_{\pm 0.2}$ | $88.4_{\pm 0.1}$ |
| CreBNN | $90.7_{\pm 0.6}$ | $88.5_{\pm 0.2}$ | $88.0_{\pm 0.2}$ | $93.5_{\pm 0.2}$ | $85.9_{\pm 0.2}$ |

CreEns depends on a hyperparameter $\alpha$, and we report their best results. The CreDE training hyperparameter is set to $0.5$, as recommended by Wang et al. (2024). Additional experimental details are provided in Appendix B.1.

**EU Quantification.** For credal classifiers, EU is estimated via the upper and lower entropy difference (e.g., (12) for CreDRO). For classical ensembles (DE & EN-DRO), EU is approximated by the mutual information following (5).

**Results.** Table 1 reports the OOD detection scores across all methods. Our proposed CreDRO consistently performs the best, indicating its effective representation of EU.

The principled derivation of a single representative probability vector from credal sets remains an open problem (Löhr et al., 2025) and is outside the scope of this work. Nevertheless, for completeness, we show that simply using the averaged probability vector $\tilde{p} \in \mathcal{K}_B$ produced by CreDRO (which yields the same prediction as DRO) outperforms classical DE, achieving higher accuracy and lower expected calibration error (ECE) (Guo et al., 2017) on the CIFAR10 test set, as shown in Table 2. Note that ECE is also defined for single-probability predictions; a principled extension of ECE to credal sets requires further investigation (Wang et al., 2024; Chau et al., 2025b). In addition, we report the test accuracy of each ensemble member for credal classifiers in Table 8 in the Appendix. The results show that the relaxation of the i.i.d. assumption during training also improves the test accuracy of the individual members of CreDRO.

*Table 2.* Test accuracy and ECE comparison using the averaged probability vector as the single prediction. Best scores are bold.

|  | Test Accuracy | ECE |
|---|---|---|
| DE | $0.9569_{\pm 0.0004}$ | $0.0051_{\pm 0.0004}$ |
| **CreDRO** | $\mathbf{0.9637}_{\pm 0.0004}$ | $\mathbf{0.0038}_{\pm 0.0008}$ |

To assess training complexity, we report the training runtime (in seconds) of credal classifiers trained on CIFAR10 with $M = 5$, measured on a single Nvidia A100-SXM4-40GB GPU. We also measure the inference time and UQ runtime (in seconds) on the CIFAR10 test set (1000 samples) using

*Table 3.* Training and inference (including UQ) time comparison in seconds for credal classifiers on the CIFAR10 dataset. Mean with standard deviation over 3 runs.

|  | Training Time | Inference Time | UQ Runtime |
|---|---|---|---|
| CreDRO | $6567.93_{\pm 35.78}$ | $1.89_{\pm 0.02}$ | $116.37_{\pm 0.30}$ |
| CreEns$_{0.0}$ | $6259.74_{\pm 15.01}$ | $2.07_{\pm 0.02}$ | $308.00_{\pm 7.93}$ |
| CreWra | $6259.74_{\pm 15.01}$ | $1.91_{\pm 0.02}$ | $123.83_{\pm 0.24}$ |
| CreRL$_{1.0}$ | $6679.69_{\pm 46.57}$ | $1.90_{\pm 0.02}$ | $133.31_{\pm 1.04}$ |
| CreDE | $6760.24_{\pm 52.81}$ | $2.03_{\pm 0.02}$ | $165.20_{\pm 0.96}$ |

the same device. The results in Table 3 show that CreDRO achieves comparable performance on these metrics. Specifically, CreDRO is lighter than CreDE in terms of training and inference, since the latter has double-sized output neurons. The higher training complexity compared to classical ensemble training (CreWra & CreEns) arises because CreDRO orders the loss of individual samples within each batch (see Algorithm 1), introducing additional computation. For UQ runtime, CreEns is the heaviest, mainly because it deploys a convex-hull credal set as representation (see Appendix G.1 for further analysis). Appendix G.2 provides additional analysis of UQ runtime when employing box credal sets.

### 4.2. Ablation Study on Ensemble Size

**Setup.** This experiment further evaluates our CreDRO under different ensemble sizes, i.e., $M \in \{5, 10, 15\}$, on the OOD detection benchmarks as in Section 4.1. All other experimental settings remain unchanged.

**Results.** Figure 2 shows that CreDRO consistently outperforms all baselines, verifying its high-quality EU representation. In addition, the performance improves with larger ensemble sizes. Detailed scores appear in Table 10 in the Appendix. Furthermore, Figure 3 shows kernel density estimates (Silverman, 2018) of EU for both ID and OOD samples. CreDRO yields notably higher EU values for OOD samples than for ID ones, qualitatively verifying its reliable EU estimation. Similar positive results are observed in kernel density plots for other ensemble sizes in Figure 8 and Figure 9 in the Appendix.

### 4.3. Ablation Study on Training Hyperparameter $\delta_G$

**Setup.** In the main experiment, we set $\delta_G$ to $0.5$ by default to reflect a balanced assessment of train-test divergence and to evaluate how this value helps our model outperform the baselines. This experiment conducts an ablation study on different values of the hyperparameter $\delta_G$ in CreDRO training, i.e., $\delta_G \in \{0.5, 0.6, 0.7, 0.8, 0.9\}$, and compare their OOD detection performance on benchmarks CIFAR10 vs SVHN, Places365, CIFAR100, FMNIST, and ImageNet. We use the setting $M = 5$ and keep all other experimental configurations unchanged, as given in Section 4.1.

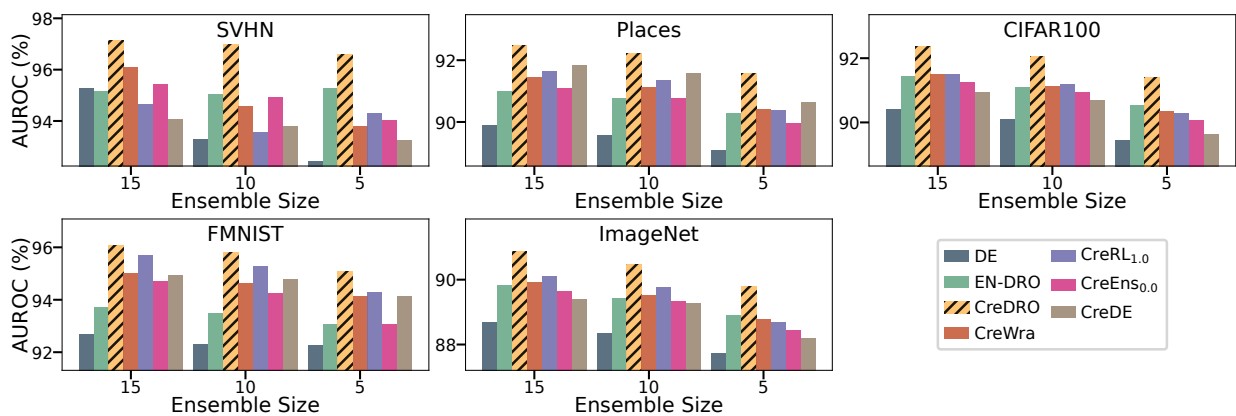

*Figure 2.* AUROC (%) for OOD detection using EU, across methods and ensemble sizes.

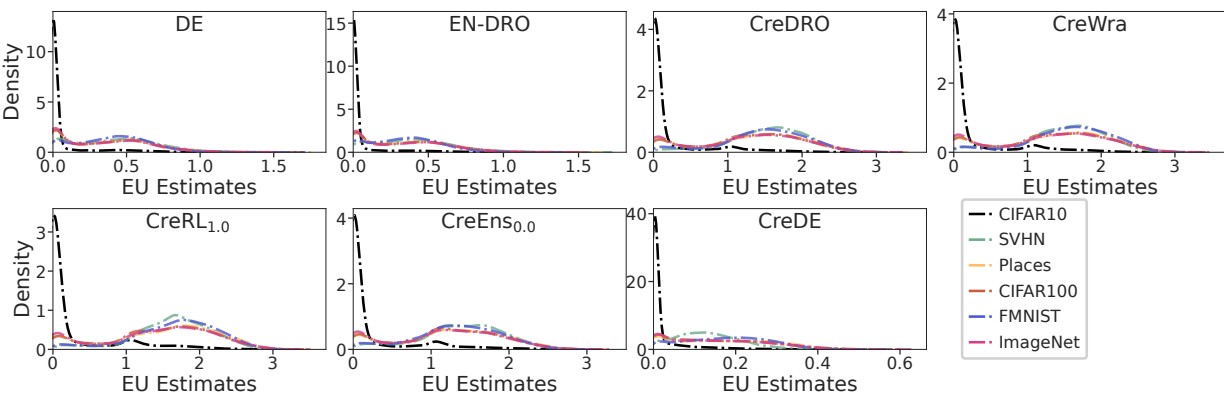

*Figure 3.* Kernel density plots of EU estimates on ID and OOD data from distinct methods. (ensemble size: $M = 5$; first-run results)

*Table 4.* AUROC score (%) of CreDRO trained with different values of $\delta_G$ for OOD detection, using CIFAR10 as the ID dataset.

|  | SVHN | Places | CIFAR100 | FMNIST | ImageNet |
|---|---|---|---|---|---|
| $\delta_G = 0.5$ | $96.6_{\pm 0.8}$ | $91.6_{\pm 0.1}$ | $91.4_{\pm 0.1}$ | $95.1_{\pm 0.1}$ | $89.8_{\pm 0.3}$ |
| $\delta_G = 0.6$ | $96.2_{\pm 0.0}$ | $91.7_{\pm 0.0}$ | $91.3_{\pm 0.1}$ | $95.1_{\pm 0.3}$ | $89.8_{\pm 0.4}$ |
| $\delta_G = 0.7$ | $95.8_{\pm 0.3}$ | $91.5_{\pm 0.1}$ | $91.2_{\pm 0.0}$ | $94.9_{\pm 0.0}$ | $89.6_{\pm 0.3}$ |
| $\delta_G = 0.8$ | $96.3_{\pm 0.4}$ | $91.6_{\pm 0.0}$ | $91.3_{\pm 0.2}$ | $95.1_{\pm 0.1}$ | $89.8_{\pm 0.2}$ |
| $\delta_G = 0.9$ | $96.1_{\pm 0.4}$ | $91.6_{\pm 0.0}$ | $91.4_{\pm 0.1}$ | $95.1_{\pm 0.3}$ | $89.8_{\pm 0.2}$ |

**Results.** Table 4 shows that CreDRO's performance is stable across different choices of $\delta_G$. This is because $\delta_G$ reflects only a single subjective belief of the model trainer about the worst-case training–test distribution divergence. In contrast, CreDRO incorporates multiple sensitivity levels to potential distributional shifts through (8), which mitigates the effect of choosing a particular $\delta_G$. Additional empirical analysis is provided in Appendix E.

### 4.4. Ablation Study on Credal Set Constructions

**Setup.** Building on our main experiment on OOD detection benchmarks in Section 4.1, we evaluate CreDRO under different credal-set construction methods, i.e., $\mathcal{K}_C$ in (11) and $\mathcal{K}_B$ in (10). We vary the ensemble sizes, i.e., $M \in \{5, 10, 15, 20\}$, while keeping all other experimental

settings the same.

*Table 5.* AUROC score (%) for OOD detection based on EU using CIFAR10 as the ID dataset, varying with different ensemble sizes. The best performance is highlighted in bold.

|  |  | SVHN | Places | CIFAR100 | FMNIST | ImageNet |
|---|---|---|---|---|---|---|
| $M = 5$ | $\mathcal{K}_B$ | $\mathbf{96.6}_{\pm 0.8}$ | $\mathbf{91.6}_{\pm 0.1}$ | $\mathbf{91.4}_{\pm 0.1}$ | $\mathbf{95.1}_{\pm 0.1}$ | $\mathbf{89.8}_{\pm 0.3}$ |
|  | $\mathcal{K}_C$ | $96.0_{\pm 0.9}$ | $91.0_{\pm 0.1}$ | $91.0_{\pm 0.1}$ | $94.1_{\pm 0.1}$ | $89.3_{\pm 0.3}$ |
| $M = 10$ | $\mathcal{K}_B$ | $\mathbf{97.0}_{\pm 0.1}$ | $\mathbf{92.2}_{\pm 0.1}$ | $\mathbf{92.1}_{\pm 0.0}$ | $\mathbf{95.8}_{\pm 0.1}$ | $\mathbf{90.5}_{\pm 0.1}$ |
|  | $\mathcal{K}_C$ | $96.6_{\pm 0.1}$ | $91.9_{\pm 0.1}$ | $91.8_{\pm 0.0}$ | $95.2_{\pm 0.2}$ | $90.2_{\pm 0.1}$ |
| $M = 15$ | $\mathcal{K}_B$ | $\mathbf{97.1}_{\pm 0.3}$ | $\mathbf{92.5}_{\pm 0.1}$ | $\mathbf{92.4}_{\pm 0.1}$ | $\mathbf{96.1}_{\pm 0.1}$ | $\mathbf{90.9}_{\pm 0.2}$ |
|  | $\mathcal{K}_C$ | $96.8_{\pm 0.4}$ | $92.2_{\pm 0.1}$ | $92.1_{\pm 0.1}$ | $95.6_{\pm 0.1}$ | $90.6_{\pm 0.2}$ |
| $M = 20$ | $\mathcal{K}_B$ | $\mathbf{97.4}_{\pm 0.1}$ | $\mathbf{92.7}_{\pm 0.1}$ | $\mathbf{92.5}_{\pm 0.1}$ | $\mathbf{96.4}_{\pm 0.0}$ | $\mathbf{91.1}_{\pm 0.1}$ |
|  | $\mathcal{K}_C$ | $97.2_{\pm 0.1}$ | $92.4_{\pm 0.1}$ | $92.3_{\pm 0.1}$ | $96.0_{\pm 0.1}$ | $90.8_{\pm 0.2}$ |

**Results.** Table 5 shows that $\mathcal{K}_B$ consistently outperforms $\mathcal{K}_C$ across all OOD detection benchmarks considered. This can be explained by the fact that OOD detection relies on the relative magnitude of EU estimates between in-distribution (ID) and OOD samples. Since $\mathcal{K}_C \subseteq \mathcal{K}_B$, using $\mathcal{K}_B$ tends to yield larger EU estimates for OOD samples, while the increase in EU estimates for ID samples remains small (as shown in Table 15 in Appendix F). Consequently, the wider EU gap between OOD and ID samples produced by $\mathcal{K}_B$

leads to improved OOD detection performance. Importantly, $\mathcal{K}_B$ does not produce overly large EU estimates on ID samples relative to $\mathcal{K}_C$, as further evidenced in Appendix F. Furthermore, since both credal sets are constructed from the same ensemble of probability vectors, the mean probability vector used for point prediction—and thus accuracy and calibration evaluation—is identical under both representations. Consequently, the two credal sets yield equivalent ECE and accuracy on ID data.

### 4.5. Robustness to Label Noise

**Setup.** In this ablation study, we investigate the robustness of CreDRO's high-loss sample upweighting strategy to label noise, which can also induce high-loss instances during training. Specifically, we inject symmetric label noise at rates of 10% and 20% into the CIFAR-10 training data while keeping the OOD test sets clean. To isolate the contribution of top-loss selection from mere sample reweighting, we introduce *CreRAM*, a variant that randomly upweights a fraction of batch samples without loss-guided selection.

**Results.** As shown in Table 6 (and Table 9 for per-member prediction accuracy in Appendix C), CreDRO consistently outperforms both CreRAM and non-DRO baselines across all noise levels, demonstrating that the performance gains stem from semantically guided selection rather than arbitrary reweighting. This robustness can be explained by the fundamentally different nature of the two loss sources. Noisy labels produce erratic and inconsistent loss signals throughout training, whereas minority-group samples exhibit systematic and structurally consistent high loss due to underrepresentation, making them more stably selected under the top-loss selection criterion. Mislabeled instances, by contrast, are unlikely to dominate this selection consistently across epochs.

### 4.6. Evaluation on Corrupted Data

**Setup.** This experiment uses OOD detection benchmarks on CIFAR10 vs CIFAR10-C (Hendrycks & Dietterich, 2019) and CIFAR100 vs CIFAR100-C (Hendrycks & Dietterich, 2019). The CIFAR10-C and CIFAR100-C datasets apply 15 types of corruptions to the CIFAR10 and CIFAR100 original test samples, respectively, with 5 severity levels for each corruption type. These benchmarks allow us to further assess the UQ performance of CreDRO under data distribution shifts.

We mainly compare CreDRO with baselines that employ the DRO concept during training, namely EN-DRO and CreDE. To broaden the range of experimental settings, we conduct the CIFAR10 vs CIFAR10-C experiment using a pre-trained ResNet50, which is then fine-tuned on the CIFAR10 training set for all methods. The input images are resized to a shape of $(224, 224, 3)$. For experiments involving CI-

*Table 6.* AUROC (%) for OOD detection ($M{=}5$). Models are trained on CIFAR-10 with symmetric label noise at varying noise rates; the clean CIFAR-10 test set is used as the ID dataset. **Black bold**: best; **Gray bold**: second best. Averaged over 3 runs.

| | SVHN | Places | CIFAR100 | FMNIST | ImageNet |
|---|---|---|---|---|---|
| Label noise level in CIFAR10 training data 10% | | | | | |
| CreRAM | $88.7_{\pm1.9}$ | $85.4_{\pm0.6}$ | $84.9_{\pm0.1}$ | $88.9_{\pm0.3}$ | $84.3_{\pm0.2}$ |
| **CreDRO** | $\mathbf{92.2}_{\pm1.3}$ | $\mathbf{87.6}_{\pm0.5}$ | $\mathbf{87.1}_{\pm0.2}$ | $91.8_{\pm0.1}$ | $\mathbf{85.9}_{\pm0.0}$ |
| CreWra | $87.4_{\pm2.0}$ | $86.5_{\pm0.3}$ | $86.2_{\pm0.2}$ | $89.2_{\pm0.3}$ | $85.0_{\pm0.1}$ |
| CreRL | $87.3_{\pm0.8}$ | $86.2_{\pm0.2}$ | $86.1_{\pm0.1}$ | $90.6_{\pm0.4}$ | $85.0_{\pm0.0}$ |
| CreEns | $88.3_{\pm0.3}$ | $85.3_{\pm0.2}$ | $85.4_{\pm0.3}$ | $87.0_{\pm0.3}$ | $84.1_{\pm0.1}$ |
| CreDE | $86.4_{\pm1.5}$ | $87.0_{\pm0.2}$ | $85.9_{\pm0.2}$ | $\mathbf{93.3}_{\pm0.9}$ | $84.9_{\pm0.2}$ |
| Label noise level in CIFAR10 training data 20% | | | | | |
| CreRAM | $83.8_{\pm1.0}$ | $79.4_{\pm0.6}$ | $78.9_{\pm0.5}$ | $81.6_{\pm1.1}$ | $78.4_{\pm0.6}$ |
| **CreDRO** | $\mathbf{88.9}_{\pm1.8}$ | $\mathbf{84.3}_{\pm0.3}$ | $\mathbf{83.6}_{\pm0.6}$ | $89.8_{\pm0.9}$ | $\mathbf{82.9}_{\pm0.5}$ |
| CreWra | $83.7_{\pm0.1}$ | $82.2_{\pm0.2}$ | $81.7_{\pm0.4}$ | $84.4_{\pm0.8}$ | $80.8_{\pm0.2}$ |
| CreRL | $81.9_{\pm3.1}$ | $82.1_{\pm0.4}$ | $81.8_{\pm0.3}$ | $86.3_{\pm1.6}$ | $80.8_{\pm0.3}$ |
| CreEns | $84.0_{\pm1.4}$ | $80.8_{\pm0.3}$ | $80.7_{\pm0.5}$ | $82.6_{\pm0.9}$ | $79.6_{\pm0.3}$ |
| CreDE | $81.0_{\pm1.1}$ | $\mathbf{84.4}_{\pm0.4}$ | $82.6_{\pm0.3}$ | $\mathbf{93.3}_{\pm0.7}$ | $81.7_{\pm0.5}$ |

FAR100, all methods are implemented on a wide residual network (WRN-28-4) backbone (Zagoruyko & Komodakis, 2016) and trained from scratch. Full implementation details are provided in Appendix B.2.

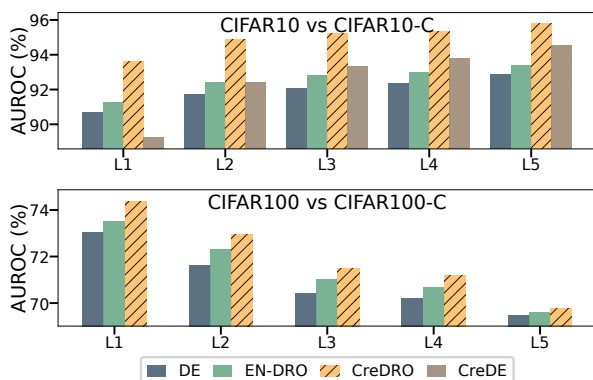

*Figure 4.* OOD detection score comparison across increasing levels of corruption. Results are averaged from 15 runs.

**Results.** Figure 4 shows the OOD detection comparison on CIFAR10 vs CIFAR10-C and CIFAR100 vs CIFAR100-C under increasing corruption intensities. The result demonstrates that our CreDRO significantly and consistently enhances EU estimation over baselines, as reflected by improved OOD detection performance across diverse dataset pairs and backbone architectures.

Note that the results of CreDE on CIFAR100 vs CIFAR100-C are missing because its generated probability intervals are too tight (see Figure 7 in the Appendix), making the optimization in (12) for calculating EU considerably difficult. For CIFAR100 vs CIFAR100-C, we did not observe a monotonic improvement in OOD detection as corruption intensity increased. This may be partly because corruption

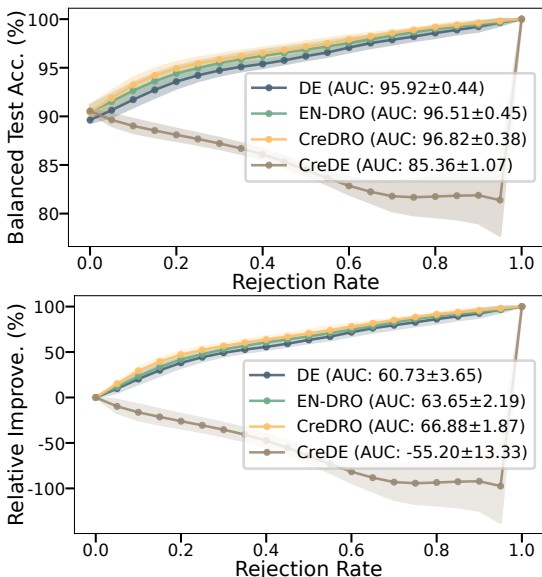

*Figure 5.* AR (top) and normalized AR (bottom) curves, along with the average AUC values. Note that the accuracy is set to 100% when the rejection rate reaches 1.0.

levels are defined from a human semantic perspective. Additionally, there is no consensus that detection performance should theoretically increase monotonically with increased corruption level.

### 4.7. Selective Classification in Medical Settings

**Setup.** In addition to OOD detection, *selective classification* is an alternative practical task for EU evaluation (Chau et al., 2025a). In this task, the accuracy–rejection (AR) curve characterizes predictive accuracy as a function of the rejection rate. Specifically, when processing a batch of instances, samples with higher EU estimates are first rejected, and accuracy is computed on the remaining test data. With reliable uncertainty estimates, the AR curve exhibits a monotonic increasing trend (Hühn & Hüllermeier, 2008; Hüllermeier et al., 2022). Besides, the area under the curve (AUC) is adopted for numerical comparison (Jaeger et al., 2023), where higher values indicate better performance.

This evaluation focuses on a real-world and large-scale binary histopathology image classification task on the Camelyon17 dataset (Bandi et al., 2018). This setting conveys a strong and realistic domain-shift scenario (Mehrtens et al., 2023), where test samples are collected from different clinics and scanned using devices that differ from those used during training. Specifically, the data from centers 0, 1, and 3, which use 3DHistech scanners, make up the ID training and validation set. The test set consists of images from centers 2 and 4, which use Philips and Hamamatsu scanners, respectively, as shown in Table 7 in the Appendix.

Our implementation follows the recipe from Mehrtens et al. (2023). Specifically, we train ResNet34-based DE, Cre-

DRO, and CreDE models and evaluate their EU quantification performance on distribution-shifted test data. The full experimental setup is detailed in Appendix B.3.

**Results.** The AR curves and their AUC values in Figure 5 (top) show that our CreDRO achieves the best performance under this realistic distribution shift, while CreDE's EU estimates appear unreliable, as accuracy decreases with higher rejection rates.

To decouple selective performance from the model's base classification accuracy, we further report normalized AR curves, where accuracy gains are measured relative to the test accuracy without rejection. Specifically, let $A(r)$ denote the accuracy at rejection rate $r$; we define the relative accuracy improvement as $\frac{A(r)-A(0)}{100\%-A(0)}$. The result in Figure 5 (bottom) further supports the strong performance of CreDRO. Additionally, we present EU estimates for correctly and incorrectly classified samples in Figure 10 in the Appendix, showing CreDRO produces noticeably higher EU values for incorrectly-classified instances, confirming its reliable EU estimation.

## 5. Conclusion and Future Work

**Conclusion.** In this paper, we propose a novel approach for quantifying epistemic uncertainty, termed CreDRO. Cre-DRO learns an ensemble of plausible models via distributionally robust optimization by varying a weight hyperparameter to simulate different levels of potential train–test distribution shift during training, corresponding to different degrees of relaxation of the i.i.d. assumption. Extensive empirical results show that ① CreDRO consistently outperforms SOTA credal classifiers and strong deep ensemble baselines on OOD detection benchmarks, and ② improves performance in selective classification in a medical setting.

**Limitation & Future Work.** In our current work, we adopt a flexible implementation of the DRO strategy for efficient batch-wise training; exploring alternative formulations within the broader DRO family may further improve performance or provide additional insights. In addition, investigating principled approaches to derive a single predictive probability from credal sets remains an important and interesting direction for future research. Finally, a rigorous theoretical analysis and empirical studies on extending our CreDRO to regression tasks (a possible road-map is provided in Appendix H) are also left for future work.

## Acknowledgement

We thank the anonymous reviewers for their valuable feedback. This work has received funding from the European Union's Horizon 2020 research and innovation program under grant agreement No. 964505 (E-pi).

## Impact Statement

This paper presents work aimed at advancing the field of uncertainty quantification in machine learning. There are many potential societal consequences of our work, none of which we feel must be specifically highlighted here.

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

# A. Extended Related Work

Credal sets have recently gained growing interest within the broader machine learning community for epistemic uncertainty (EU) quantification, even prior to their application in deep learning. Relevant studies include (Zaffalon, 2002; Corani & Zaffalon, 2008; Corani et al., 2012; Mauá et al., 2017; Hüllermeier & Waegeman, 2021; Shaker & Hüllermeier, 2021a; Sale et al., 2023; Wang et al., 2026). A key benefit of credal sets is their unification of set-based and distribution-based concepts into a single framework. This makes them a more natural way to represent epistemic uncertainty than modeling with individual probability distributions (Corani et al., 2012; Hüllermeier & Waegeman, 2021). For instance, one can argue that sets better capture ignorance as a lack of knowledge (Dubois et al., 2002), since distributions inherently rely on extra assumptions beyond simply distinguishing between plausible and implausible candidates (Löhr et al., 2025). For more motivations for working with credal sets for uncertainty quantification, please refer to Appendix A of Caprio et al. (2024).

In deep learning, a recent credal wrapper (CreWra) approach (Wang et al., 2025b) formulates credal set predictions from the multiple softmax predictions from members of a deep ensemble (DE) (Lakshminarayanan et al., 2017) trained with different random initializations. A credal ensembling (CreEns) (Nguyen et al., 2025) acts as an alternative post hoc extension to a classical ensemble that also enables discarding outlier probabilities, preventing credal sets from becoming too large. Credal predictions based on relative likelihood (CreRL) (Löhr et al., 2025) proposes a Tobias scheme to increase diversity in ensemble initialization and a selection process based on a predefined threshold ($\alpha$ cut) to identify plausible members of an ensemble by their relative likelihood. Although these credal predictors improve EU estimation, they mainly focus on classification and capture EU stemming from ensemble disagreement due to random initializations. They fail to adequately reflect ignorance caused by potential shifts between training and testing distributions—a critical challenge in practical applications. Alternatively, a recent approach, credal deep ensemble (CreDE) (also used as a baseline in our comparison), directly trains neural networks to predict probability intervals by outputting a lower and an upper probability for each class (Wang et al., 2024). Although also incorporating the distributionally robust optimization (DRO) concept in learning the lower probability bounds, the training scheme between CreDRO and CreDE is quite different, as we detailed in Section 3.1. In addition, our extensive experiments in Section 4 demonstrate that our CreDRO consistently outperforms CreDE. From the practical perspectives, these aforementioned methods present distinct constraints: CreDE requires modifying the model's last layer; CreWra and CreEns operate solely as post-hoc processors; and CreRL relies on a maximum-likelihood estimator as a reference model and lacks a clear criterion for choosing $\alpha$. These limitations restrict broader applicability in practice.

Beyond these recent advancements, other methods exist: credal Bayesian deep learning (CreBNN) (Caprio et al., 2024), which we also employ as a baseline for comparison in Section 4. CreBNN represents weights and predictions as credal sets and has demonstrated robustness; however, the setting of credal set prior still relies on random initialization and its computational cost is comparable to that of Bayesian neural network ensembles, significantly limiting practical applicability. Additionally, Wang et al. (2025c) introduces credal-set interval neural networks, which construct credal sets from probability intervals derived from the deterministic interval outputs of interval neural networks. This approach offers a unique capability for uncertainty quantification (UQ) with interval input data and achieves EU quantification performance comparable to deep ensembles, particularly on smaller datasets, with greater efficiency. Nonetheless, its computational cost hinders scalability. Since improving EU quality is one of our primary focuses, its comparable performance with deep ensembles, coupled with the scalability constraint, led us to exclude it from our baselines. Moreover, another approach generates credal predictions using belief functions defined by a random-set output (Manchingal et al., 2025). Training such a model requires a budgeting process to select the most relevant class and to convert the original one-hot labels into belief-function form. This deviates from our setting, e.g., learning is based on the given original data. Thus, we do not include it as a baseline.

Uncertainty quantification for credal sets remains an active area of research (Hüllermeier & Waegeman, 2021; Sale et al., 2023). In classification problems, the notions of generalized entropy (Abellán et al., 2006) and the generalized Hartley (GH) measure (Abellán & Moral, 2000; Hüllermeier et al., 2022) have been introduced. However, practical application of the GH measure is often hindered by the computational complexity of solving constrained optimization problems over $2^C$ subsets (Hüllermeier & Waegeman, 2021; Hüllermeier et al., 2022), which becomes especially challenging when the number of classes $C$ is large (e.g., $C = 100$). Since intervals represent credal sets in the special binary case, (Hüllermeier et al., 2022) recently proposed using the width of the probability interval as an EU measure. Their results indicate that this measure offers improved theoretical properties and empirical performance compared to generalized entropy (Abellán et al., 2006) and the GH measure in the binary case. More recently, an imprecise probability metric framework proposes the maximum mean imprecision measure based on the total variance distance to quantify credal epistemic uncertainty (Chau et al., 2025a). We include generalized entropy in multiclass classification primarily to enable fair comparison, as it is widely used among state-of-the-art credal classifiers.

# B. Experiment Details

This section details the experimental setup for our evaluation. The datasets are drawn from existing literature. The core implementation code for running and analyzing the experiments is publicly available at `https://github.com/Kaizh eng-WANG/Learning-Credal-Ensembles-via-Distributionally-Robust-Optimization`.

## B.1. Comparison with SOTA credal classifiers and DE baseline

In this experiment, we follow the experimental setup of Löhr et al. (Löhr et al., 2025). The training procedures for the baselines, including CreBNN (Caprio et al., 2024), CreDE (Wang et al., 2024), CreWra (Wang et al., 2025b), CreEns (Nguyen et al., 2025), and CreRL (Löhr et al., 2025), are reproduced using the official GitHub repository `https://github .com/timoverse/credal-prediction-relative-likelihood`. We then train CreDRO with the training hyperparameter set to $\delta_G = 0.5$, under the same settings to ensure a fair comparison.

Specifically, we use the PyTorch ResNet18 implementation and hyperparameters provided by `https://github.com/k uangliu/pytorch-cifar`. This model is optimized for CIFAR10 and is trained from scratch without any pretraining on ImageNet. The model is trained for 200 epochs with an initial learning rate of 0.1. We use SGD as the optimizer with a weight decay of 0.0005 and a batch size of 128. A cosine annealing learning rate scheduler is applied to gradually decay the learning rate during training. All experiments are conducted on a single Nvidia A100-SXM4-80GB GPU.

**Expected Calibration Error (ECE) Evaluation.** Within the ECE framework, a single probabilistic prediction is considered well calibrated if predictions made with a confidence of 80% are correct in roughly 80% of cases. To compute ECE, model predictions are partitioned into a fixed number $G$ of confidence bins $B_g$, each covering an equal interval of confidence scores. The ECE metric is obtained by aggregating the weighted absolute differences between the empirical accuracy and the average confidence in each bin (Mehrtens et al., 2023):

$$\sum_{g=1}^{G} \frac{|B_g|}{n} \left| \mathrm{acc}(B_g) - \mathrm{conf}(B_g) \right|, \tag{13}$$

where $|B_g|$ denotes the number of samples assigned to the $g$-th bin, and $n$ represents the total number of samples. Following common practice, we set the number of bins to the default value of $G = 10$ in our experiments.

## B.2. Evaluation on Corrupted Data

To broaden the range of experimental settings, we conduct the CIFAR10 vs CIFAR10-C experiment using a pre-trained ResNet50, which is then fine-tuned on the CIFAR10 training set for all methods, in the framework of TensorFlow. The input images are resized to $(224, 224, 3)$. For experiments involving CIFAR100, a wide residual network (WRN-28-4) (Zagoruyko & Komodakis, 2016) is used as the backbone and trained from scratch for all methods.

Regarding fine-tuning on the pre-trained ResNet50 using CIFAR10, standard data augmentation is applied uniformly across all methods to improve data quality and enhance training performance. The training batch size is set as 128. The standard data split is applied. The Adam optimizer is employed, with a learning rate scheduler set at 0.001 and reduced to 0.0001 during the final 5 training epochs. All models are trained using a single NVIDIA Tesla P100 SXM2 GPU for 20 epochs under different random seeds.

For scratch training on WRN-28-4 using CIFAR100, we use a learning rate schedule with linear warm-up followed by cosine annealing to improve training stability and performance, and train all methods for 200 epochs. Specifically, the learning rate is linearly increased from $10^{-3}$ to 0.1 during the first 5 epochs and then decayed to $10^{-6}$ using a cosine schedule over the remaining epochs. We use SGD with learning rate 0.1, momentum 0.9, and weight decay set to $5 \times 10^{-6}$. The other configurations, such as data split, batch size, data augmentation, and training hardware, remain unchanged.

## B.3. Selective Classification in Medical Settings

Building on the work in (Mehrtens et al., 2023), this case study uses the Camelyon17 (Bandi et al., 2018) histopathological dataset for binary classification into Tumor, Non-Tumor. The dataset contains whole slide images (WSIs) of breast lymph node tissue with lesion-level annotations highlighting metastatic areas. Camelyon17 includes 50 WSIs collected from five medical centers in the Netherlands, captured using three different scanner types. An example WSI is shown in Figure 6.

To convey a strong domain shift scenario (Mehrtens et al., 2023), the dataset is split so that the test set only contains images from scanners not present in the in-distribution (ID) training data, introducing an additional technological variation. Specifically, WSIs from centers 0, 1, and 3, which use 3DHistech scanners, make up the ID training and validation set. The test set consists of images from centers 2 and 4, which use Philips and Hamamatsu scanners, respectively. The detailed data split is summarized in Table 7.

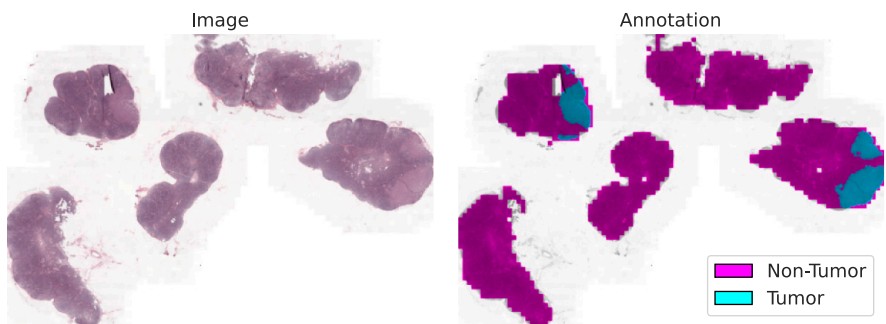

*Figure 6.* An example of the whole slide image (referring to node 2 of patient 017 in the Camelyon17 dataset) with ground-truth annotations.

*Table 7.* Camelyon17 medical tile image instances data split applied in the experiment, where the input shape is $(3, 224, 224)$.

|                | Train      | Validation | Test                |
| -------------- | ---------- | ---------- | ------------------- |
| Clinic centers | 0, 1, 3    | 0, 1, 3    | 2, 4                |
| Scanner        | 3D Histech | 3D Histech | Philips, Hamamatsu  |
| # Instances    | 383406     | 110561     | 255968              |

For the training configurations, we follow the official GitHub repository `https://github.com/DBO-DKFZ/uncertainty-benchmark` to train the deep ensemble (DE) model ($M = 5$), using the approach outlined in Mehrtens et al. (2023). We then adjust the training of CreDRO and CreDE by setting the training hyperparameter to 0.5, as suggested in the original work, to ensure a fair comparison. The neural network backbone is ResNet34, trained from scratch on a single NVIDIA Tesla P100 SXM2 GPU. All other settings, including data augmentation, optimizer, learning rate, and others, follow the default configurations of the repository. Considering mainly the computational cost, we conduct each experiment over 5 runs.

Note that, in the special case of binary classification, the credal set reduces to a single probability interval denoted as $[\underline{p}, \overline{p}]$. (Hüllermeier et al., 2022) have recently proposed using the width of this interval as an EU measure, i.e., $\overline{p} - \underline{p}$, which shows better theoretical properties and empirical performance than generalized entropy (Abellán et al., 2006). For this reason, we adopt this measure in our experiments.

## C. Additional Experiment Results

*Table 8.* Test accuracy of each ensemble member in credal classifiers on the CIFAR10 test set, providing insight into their individual performance. For CreDE, whose ensemble members directly predict probability intervals, we apply their intersection probability technique to obtain pointwise predictions (Wang et al., 2025b), following the setting of Löhr et al. (2025). The results show that incorporating DRO during training also improves the test accuracy of the individual members of CreDRO.

| Member Index | CreWra | **CreDRO** | CreEns | CreBNN | CreDE | CreRL$_{1.0}$ |
|---|---|---|---|---|---|---|
| 1 | $0.942_{\pm 0.001}$ | $0.954_{\pm 0.001}$ | $0.955_{\pm 0.000}$ | $0.875_{\pm 0.004}$ | $0.941_{\pm 0.000}$ | $0.943_{\pm 0.001}$ |
| 2 | $0.944_{\pm 0.001}$ | $0.953_{\pm 0.002}$ | $0.954_{\pm 0.000}$ | $0.877_{\pm 0.005}$ | $0.943_{\pm 0.000}$ | $0.934_{\pm 0.001}$ |
| 3 | $0.944_{\pm 0.001}$ | $0.953_{\pm 0.001}$ | $0.952_{\pm 0.001}$ | $0.867_{\pm 0.015}$ | $0.944_{\pm 0.001}$ | $0.934_{\pm 0.002}$ |
| 4 | $0.943_{\pm 0.002}$ | $0.954_{\pm 0.003}$ | $0.952_{\pm 0.000}$ | $0.881_{\pm 0.003}$ | $0.942_{\pm 0.001}$ | $0.936_{\pm 0.001}$ |
| 5 | $0.943_{\pm 0.001}$ | $0.953_{\pm 0.000}$ | $0.952_{\pm 0.001}$ | $0.872_{\pm 0.004}$ | $0.942_{\pm 0.001}$ | $0.935_{\pm 0.002}$ |
| 6 | $0.943_{\pm 0.000}$ | $0.954_{\pm 0.001}$ | $0.952_{\pm 0.000}$ | $0.869_{\pm 0.006}$ | $0.942_{\pm 0.002}$ | $0.934_{\pm 0.002}$ |
| 7 | $0.941_{\pm 0.001}$ | $0.954_{\pm 0.001}$ | $0.952_{\pm 0.001}$ | $0.877_{\pm 0.001}$ | $0.943_{\pm 0.001}$ | $0.936_{\pm 0.001}$ |
| 8 | $0.942_{\pm 0.001}$ | $0.955_{\pm 0.000}$ | $0.951_{\pm 0.001}$ | $0.880_{\pm 0.006}$ | $0.943_{\pm 0.002}$ | $0.935_{\pm 0.001}$ |
| 9 | $0.943_{\pm 0.001}$ | $0.953_{\pm 0.001}$ | $0.951_{\pm 0.000}$ | $0.879_{\pm 0.005}$ | $0.944_{\pm 0.001}$ | $0.936_{\pm 0.002}$ |
| 10 | $0.943_{\pm 0.000}$ | $0.954_{\pm 0.001}$ | $0.953_{\pm 0.000}$ | $0.873_{\pm 0.007}$ | $0.943_{\pm 0.003}$ | $0.936_{\pm 0.001}$ |
| 11 | $0.942_{\pm 0.001}$ | $0.954_{\pm 0.002}$ | $0.953_{\pm 0.001}$ | $0.882_{\pm 0.004}$ | $0.943_{\pm 0.000}$ | $0.936_{\pm 0.001}$ |
| 12 | $0.943_{\pm 0.000}$ | $0.953_{\pm 0.003}$ | $0.953_{\pm 0.000}$ | $0.879_{\pm 0.003}$ | $0.942_{\pm 0.000}$ | $0.934_{\pm 0.002}$ |
| 13 | $0.943_{\pm 0.000}$ | $0.955_{\pm 0.001}$ | $0.953_{\pm 0.001}$ | $0.874_{\pm 0.000}$ | $0.943_{\pm 0.002}$ | $0.937_{\pm 0.002}$ |
| 14 | $0.945_{\pm 0.000}$ | $0.954_{\pm 0.001}$ | $0.952_{\pm 0.001}$ | $0.872_{\pm 0.006}$ | $0.941_{\pm 0.001}$ | $0.934_{\pm 0.002}$ |
| 15 | $0.942_{\pm 0.002}$ | $0.952_{\pm 0.001}$ | $0.948_{\pm 0.000}$ | $0.856_{\pm 0.019}$ | $0.942_{\pm 0.001}$ | $0.933_{\pm 0.002}$ |
| 16 | $0.942_{\pm 0.000}$ | $0.953_{\pm 0.002}$ | $0.941_{\pm 0.000}$ | $0.870_{\pm 0.002}$ | $0.942_{\pm 0.001}$ | $0.936_{\pm 0.002}$ |
| 17 | $0.942_{\pm 0.001}$ | $0.955_{\pm 0.000}$ | $0.930_{\pm 0.001}$ | $0.854_{\pm 0.035}$ | $0.942_{\pm 0.002}$ | $0.935_{\pm 0.001}$ |
| 18 | $0.944_{\pm 0.001}$ | $0.954_{\pm 0.001}$ | $0.921_{\pm 0.001}$ | $0.872_{\pm 0.007}$ | $0.943_{\pm 0.002}$ | $0.934_{\pm 0.001}$ |
| 19 | $0.943_{\pm 0.001}$ | $0.954_{\pm 0.001}$ | $0.906_{\pm 0.002}$ | $0.873_{\pm 0.001}$ | $0.942_{\pm 0.001}$ | $0.936_{\pm 0.001}$ |
| 20 | $0.942_{\pm 0.001}$ | $0.956_{\pm 0.001}$ | $0.872_{\pm 0.002}$ | $0.860_{\pm 0.012}$ | $0.942_{\pm 0.001}$ | $0.935_{\pm 0.002}$ |
| Average | 0.9428 | **0.95385** | 0.94265 | 0.8721 | 0.94245 | 0.93545 |

*Table 9.* Test accuracy of each ensemble member in credal classifiers on the CIFAR10 test set, providing insight into their individual performance. For CreDE, whose ensemble members directly predict probability intervals, we apply their intersection probability technique to obtain pointwise predictions, following the setting of *Löhr et al. (2025)*. Our CreDRO and the CreDE baseline incorporate the DRO training strategy. The results show that incorporating DRO during training also improves the test accuracy of the individual members of CreDRO. **black bold**: best; **gray bold**: second best. Results are averaged over 3 runs.

| Member Index | CreRAM | **CreDRO** | CreWra | CreRL | CreEns | CreDE |
|---|---|---|---|---|---|---|
| | | | Label noise level in CIFAR10 training data 10% | | | |
| 1 | $0.888_{\pm 0.002}$ | $0.904_{\pm 0.001}$ | $0.898_{\pm 0.001}$ | $0.898_{\pm 0.003}$ | $0.921_{\pm 0.001}$ | $0.897_{\pm 0.001}$ |
| 2 | $0.892_{\pm 0.002}$ | $0.904_{\pm 0.002}$ | $0.896_{\pm 0.003}$ | $0.884_{\pm 0.002}$ | $0.919_{\pm 0.002}$ | $0.896_{\pm 0.002}$ |
| 3 | $0.896_{\pm 0.002}$ | $0.905_{\pm 0.001}$ | $0.896_{\pm 0.002}$ | $0.876_{\pm 0.003}$ | $0.921_{\pm 0.001}$ | $0.900_{\pm 0.001}$ |
| 4 | $0.899_{\pm 0.001}$ | $0.901_{\pm 0.002}$ | $0.897_{\pm 0.003}$ | $0.883_{\pm 0.001}$ | $0.905_{\pm 0.000}$ | $0.900_{\pm 0.001}$ |
| 5 | $0.904_{\pm 0.002}$ | $0.902_{\pm 0.001}$ | $0.896_{\pm 0.002}$ | $0.884_{\pm 0.005}$ | $0.823_{\pm 0.002}$ | $0.896_{\pm 0.003}$ |
| Average | $0.896_{\pm 0.001}$ | **$0.903_{\pm 0.001}$** | $0.896_{\pm 0.002}$ | $0.885_{\pm 0.003}$ | $0.898_{\pm 0.001}$ | $0.898_{\pm 0.002}$ |
| | | | Label noise level in CIFAR10 training data 20% | | | |
| 1 | $0.813_{\pm 0.002}$ | $0.854_{\pm 0.002}$ | $0.846_{\pm 0.005}$ | $0.842_{\pm 0.003}$ | $0.889_{\pm 0.003}$ | $0.849_{\pm 0.003}$ |
| 2 | $0.816_{\pm 0.004}$ | $0.850_{\pm 0.001}$ | $0.841_{\pm 0.004}$ | $0.823_{\pm 0.005}$ | $0.885_{\pm 0.002}$ | $0.851_{\pm 0.002}$ |
| 3 | $0.827_{\pm 0.000}$ | $0.851_{\pm 0.004}$ | $0.844_{\pm 0.004}$ | $0.823_{\pm 0.007}$ | $0.886_{\pm 0.002}$ | $0.853_{\pm 0.003}$ |
| 4 | $0.835_{\pm 0.001}$ | $0.845_{\pm 0.002}$ | $0.845_{\pm 0.004}$ | $0.823_{\pm 0.006}$ | $0.850_{\pm 0.002}$ | $0.850_{\pm 0.005}$ |
| 5 | $0.841_{\pm 0.000}$ | $0.838_{\pm 0.001}$ | $0.843_{\pm 0.000}$ | $0.824_{\pm 0.001}$ | $0.701_{\pm 0.004}$ | $0.850_{\pm 0.003}$ |
| Average | $0.827_{\pm 0.001}$ | $0.847_{\pm 0.002}$ | $0.844_{\pm 0.003}$ | $0.827_{\pm 0.004}$ | $0.842_{\pm 0.003}$ | **$0.850_{\pm 0.003}$** |

*Table 10.* AUROC score (%) for OOD detection based on EU using CIFAR10 as the ID dataset. The best performance is highlighted in bold. Results vary with different ensemble sizes, showing that CreDRO consistently outperforms all baselines, further confirming its high-quality EU representation.

| | SVHN | Places | CIFAR100 | FMNIST | ImageNet |
|---|---|---|---|---|---|
| | | Ensemble Number $M = 5$ | | | |
| DE | $92.4_{\pm 0.3}$ | $89.1_{\pm 0.2}$ | $89.4_{\pm 0.2}$ | $92.3_{\pm 0.2}$ | $87.7_{\pm 0.2}$ |
| **EN-DRO** | $95.3_{\pm 0.9}$ | $90.3_{\pm 0.1}$ | $90.5_{\pm 0.0}$ | $93.1_{\pm 0.2}$ | $88.9_{\pm 0.1}$ |
| **CreDRO** | $\mathbf{96.6}_{\pm 0.8}$ | $\mathbf{91.6}_{\pm 0.1}$ | $\mathbf{91.4}_{\pm 0.1}$ | $\mathbf{95.1}_{\pm 0.1}$ | $\mathbf{89.8}_{\pm 0.3}$ |
| CreWra | $93.8_{\pm 0.3}$ | $90.4_{\pm 0.2}$ | $90.4_{\pm 0.1}$ | $94.1_{\pm 0.2}$ | $88.8_{\pm 0.2}$ |
| CreRL$_{1.0}$ | $94.3_{\pm 0.3}$ | $90.4_{\pm 0.1}$ | $90.3_{\pm 0.1}$ | $94.3_{\pm 0.2}$ | $88.7_{\pm 0.1}$ |
| CreEns$_{0.0}$ | $94.0_{\pm 0.6}$ | $89.9_{\pm 0.1}$ | $90.1_{\pm 0.0}$ | $93.1_{\pm 0.1}$ | $88.5_{\pm 0.2}$ |
| CreDE | $93.2_{\pm 0.6}$ | $90.6_{\pm 0.2}$ | $89.6_{\pm 0.2}$ | $94.1_{\pm 0.3}$ | $88.2_{\pm 0.3}$ |
| | | Ensemble Number $M = 10$ | | | |
| DE | $93.3_{\pm 0.5}$ | $89.6_{\pm 0.2}$ | $90.1_{\pm 0.1}$ | $92.3_{\pm 0.4}$ | $88.4_{\pm 0.1}$ |
| **EN-DRO** | $95.1_{\pm 0.1}$ | $90.8_{\pm 0.1}$ | $91.1_{\pm 0.0}$ | $93.5_{\pm 0.2}$ | $89.4_{\pm 0.1}$ |
| **CreDRO** | $\mathbf{97.0}_{\pm 0.1}$ | $\mathbf{92.2}_{\pm 0.1}$ | $\mathbf{92.1}_{\pm 0.0}$ | $\mathbf{95.8}_{\pm 0.1}$ | $\mathbf{90.5}_{\pm 0.1}$ |
| CreWra | $94.6_{\pm 0.3}$ | $91.1_{\pm 0.2}$ | $91.1_{\pm 0.1}$ | $94.6_{\pm 0.3}$ | $89.5_{\pm 0.1}$ |
| CreRL$_{1.0}$ | $93.6_{\pm 0.3}$ | $91.3_{\pm 0.2}$ | $91.2_{\pm 0.1}$ | $95.3_{\pm 0.2}$ | $89.8_{\pm 0.0}$ |
| CreEns$_{0.0}$ | $94.9_{\pm 0.7}$ | $90.8_{\pm 0.1}$ | $90.9_{\pm 0.1}$ | $94.3_{\pm 0.2}$ | $89.3_{\pm 0.1}$ |
| CreDE | $93.8_{\pm 0.3}$ | $91.6_{\pm 0.2}$ | $90.7_{\pm 0.1}$ | $94.8_{\pm 0.2}$ | $89.3_{\pm 0.1}$ |
| | | Ensemble Number $M = 15$ | | | |
| DE | $95.3_{\pm 0.3}$ | $89.9_{\pm 0.2}$ | $90.4_{\pm 0.0}$ | $92.7_{\pm 0.2}$ | $88.7_{\pm 0.1}$ |
| **EN-DRO** | $95.2_{\pm 0.6}$ | $91.0_{\pm 0.1}$ | $91.4_{\pm 0.1}$ | $93.7_{\pm 0.1}$ | $89.8_{\pm 0.1}$ |
| **CreDRO** | $\mathbf{97.1}_{\pm 0.3}$ | $\mathbf{92.5}_{\pm 0.1}$ | $\mathbf{92.4}_{\pm 0.1}$ | $\mathbf{96.1}_{\pm 0.1}$ | $\mathbf{90.9}_{\pm 0.2}$ |
| CreWra | $96.1_{\pm 0.2}$ | $91.5_{\pm 0.2}$ | $91.5_{\pm 0.1}$ | $95.0_{\pm 0.1}$ | $89.9_{\pm 0.1}$ |
| CreRL$_{1.0}$ | $94.6_{\pm 0.3}$ | $91.7_{\pm 0.1}$ | $91.5_{\pm 0.1}$ | $95.7_{\pm 0.2}$ | $90.1_{\pm 0.1}$ |
| CreEns$_{0.0}$ | $95.4_{\pm 0.3}$ | $91.1_{\pm 0.2}$ | $91.2_{\pm 0.1}$ | $94.7_{\pm 0.1}$ | $89.7_{\pm 0.1}$ |
| CreDE | $94.1_{\pm 0.4}$ | $91.8_{\pm 0.1}$ | $90.9_{\pm 0.1}$ | $95.0_{\pm 0.1}$ | $89.4_{\pm 0.2}$ |
| | | Ensemble Number $M = 20$ | | | |
| DE | $94.8_{\pm 0.3}$ | $90.0_{\pm 0.2}$ | $90.6_{\pm 0.0}$ | $92.9_{\pm 0.3}$ | $88.9_{\pm 0.1}$ |
| **EN-DRO** | $95.7_{\pm 0.0}$ | $91.1_{\pm 0.1}$ | $91.6_{\pm 0.1}$ | $94.0_{\pm 0.1}$ | $90.0_{\pm 0.1}$ |
| **CreDRO** | $\mathbf{97.4}_{\pm 0.1}$ | $\mathbf{92.7}_{\pm 0.1}$ | $\mathbf{92.5}_{\pm 0.1}$ | $\mathbf{96.4}_{\pm 0.0}$ | $\mathbf{91.1}_{\pm 0.1}$ |
| CreWra | $95.7_{\pm 0.3}$ | $91.6_{\pm 0.1}$ | $91.6_{\pm 0.0}$ | $95.2_{\pm 0.0}$ | $89.0_{\pm 0.1}$ |
| CreRL$_{1.0}$ | $94.8_{\pm 0.3}$ | $91.8_{\pm 0.2}$ | $91.6_{\pm 0.1}$ | $95.7_{\pm 0.2}$ | $88.9_{\pm 0.2}$ |
| CreEns$_{0.0}$ | $95.5_{\pm 0.1}$ | $91.3_{\pm 0.0}$ | $91.4_{\pm 0.1}$ | $94.9_{\pm 0.1}$ | $88.8_{\pm 0.0}$ |
| CreDE | $94.3_{\pm 0.3}$ | $91.8_{\pm 0.0}$ | $91.2_{\pm 0.0}$ | $95.1_{\pm 0.2}$ | $88.4_{\pm 0.1}$ |

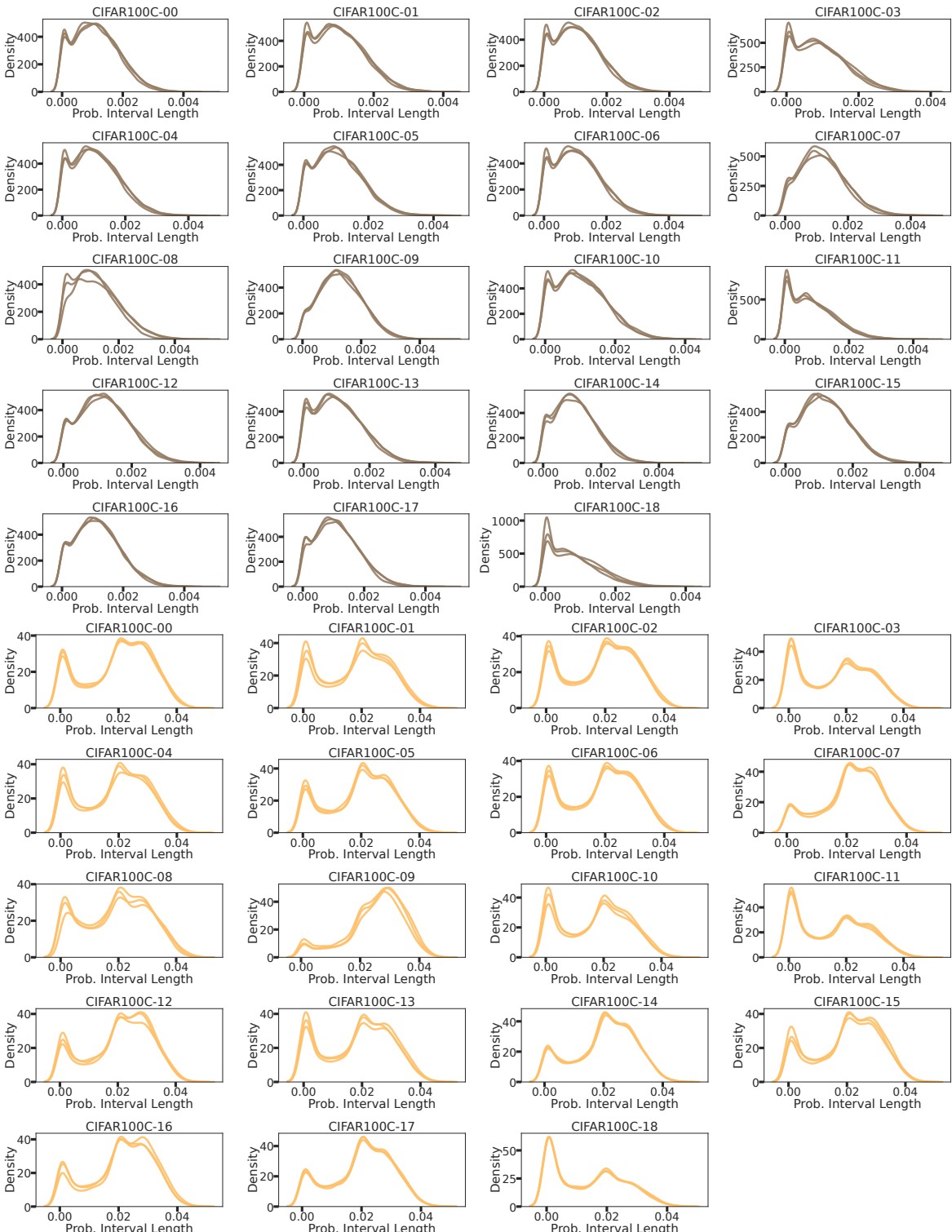

*Figure 7.* Kernel density plots of the averaged probability interval length (PIL) over classes on CIFAR100-C datasets with 19 types of corruptions: CreDE (top) and CreDRO (bottom). The averaged PIL for an input instance is computed as PIL $= \frac{1}{100} \sum_{k=1}^{100} \overline{p}_k - \underline{p}_k$. The results show that the PIL produced by CreDE is approximately 10 times tighter than that of CreDRO, which makes solving the optimization problem in (12) for EU estimation considerably more difficult. This is because, as the intervals become tighter (i.e., narrower bounds), the feasible region shrinks, often forcing the optimizer to search within a smaller, more constrained domain that frequently lies near or on the boundaries. This typically leads to more active constraints, worsened problem conditioning, and an increase in the number of iterations required for convergence.

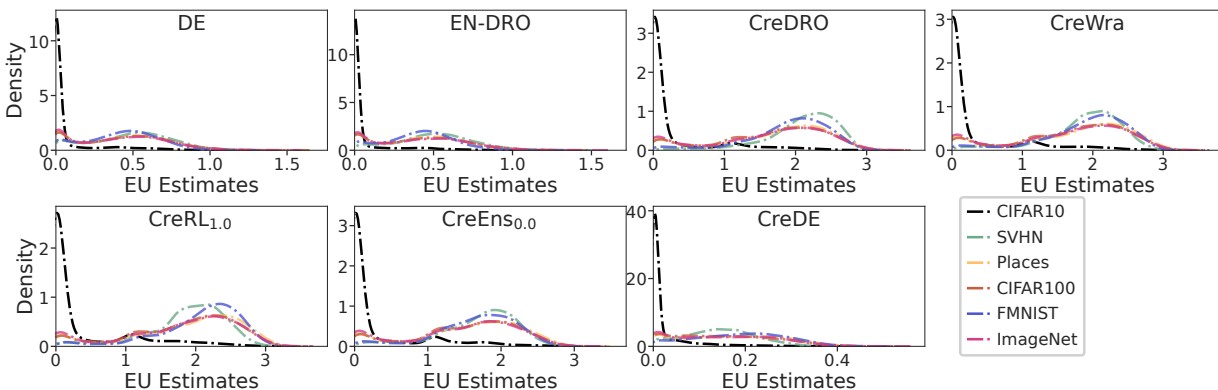

*Figure 8.* Kernel density plots of EU estimates on ID and OOD data from distinct methods. (ensemble size: $M = 10$; first-run results)

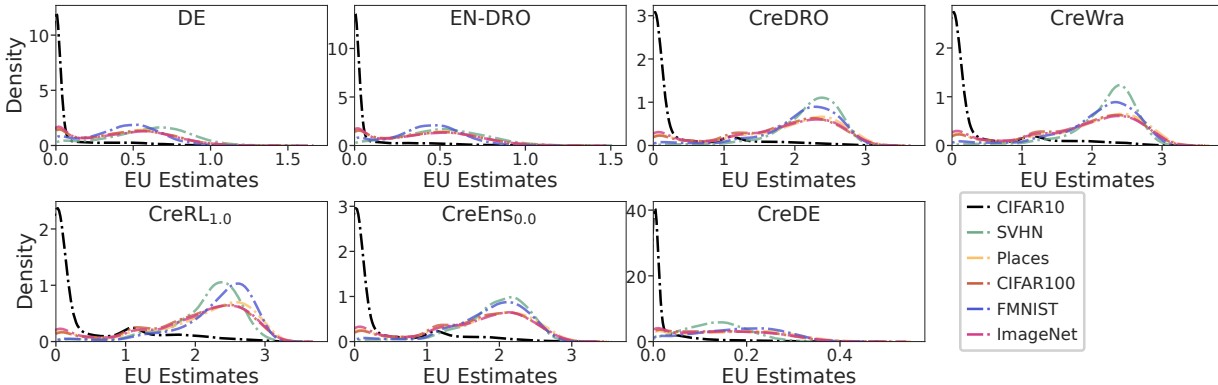

*Figure 9.* Kernel density plots of EU estimates on ID and OOD data from distinct methods. (ensemble size: $M = 15$; first-run results)

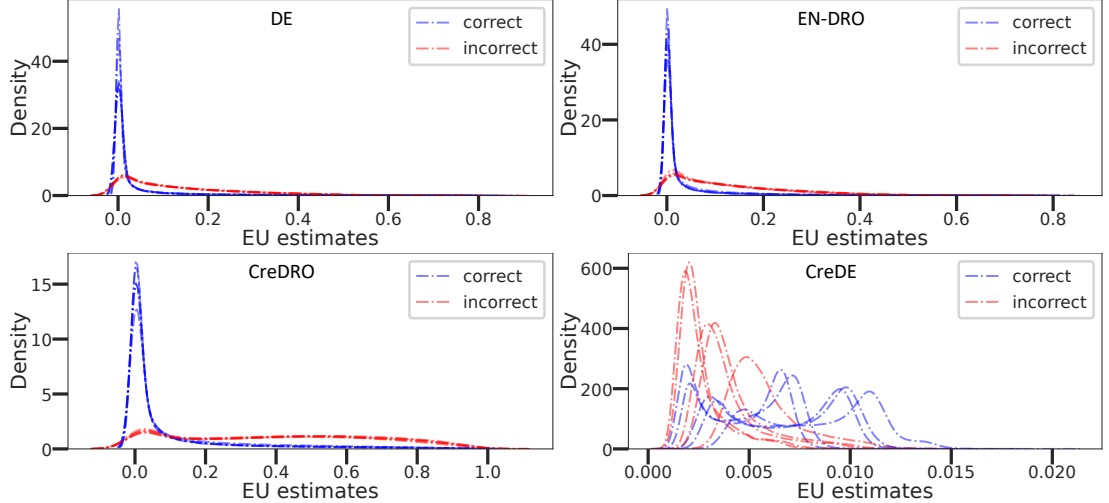

*Figure 10.* Kernel density plots of EU estimates on correctly and incorrectly classified medical test samples, across different classifiers. The plots show that CreDRO produces noticeably higher EU values for incorrectly classified samples than for correctly classified ones, qualitatively confirming its reliable EU estimation. In contrast, CreDE tends to be overconfident (overall smaller EU values) and struggles to distinguish between incorrectly and correctly classified samples.

## D. Ablation Study on Interpolation Choices

In this ablation study, we compare three interpolation strategies for $[\delta_G, 1]$ in CreDRO training: uniform as in (8), exponential-like (concentrated near $\delta_G$), and logarithmic-like (concentrated near 1), instantiated respectively as

$$\delta_i = (1 - \delta_G) \cdot \left(\frac{i - 1}{M - 1}\right)^2 + \delta_G$$

and

$$\delta_i = (1 - \delta_G) \cdot \left(\frac{i - 1}{M - 1}\right)^{0.5} + \delta_G.$$

Table 11 reports the AUROC (%) for OOD detection using CreDRO ($M = 5$) with EU on CIFAR-10 as the ID dataset. The results show that OOD detection performance is nearly identical across all three strategies, confirming CreDRO's robustness to the uniform design choice.

*Table 11.* AUROC (%) for OOD detection using CreDRO ($M = 5, \delta_G = 0.5$) with EU on CIFAR10 as the ID dataset, comparing different interpolation strategies. Results are averaged over 3 runs.

|  | SVHN | Places | CIFAR100 | FMNIST | ImageNet |
|---|---|---|---|---|---|
| Uniform interpolation | $96.6_{\pm 0.8}$ | $91.6_{\pm 0.1}$ | $91.4_{\pm 0.1}$ | $95.1_{\pm 0.1}$ | $89.8_{\pm 0.3}$ |
| Exponential-like interpolation | $96.6_{\pm 0.5}$ | $91.6_{\pm 0.0}$ | $91.4_{\pm 0.1}$ | $95.4_{\pm 0.2}$ | $89.6_{\pm 0.2}$ |
| Logarithmic-like interpolation | $96.4_{\pm 0.3}$ | $91.6_{\pm 0.1}$ | $91.4_{\pm 0.1}$ | $95.1_{\pm 0.1}$ | $89.7_{\pm 0.3}$ |

## E. Ablation Study on Training Hyperparameter $\delta_G$

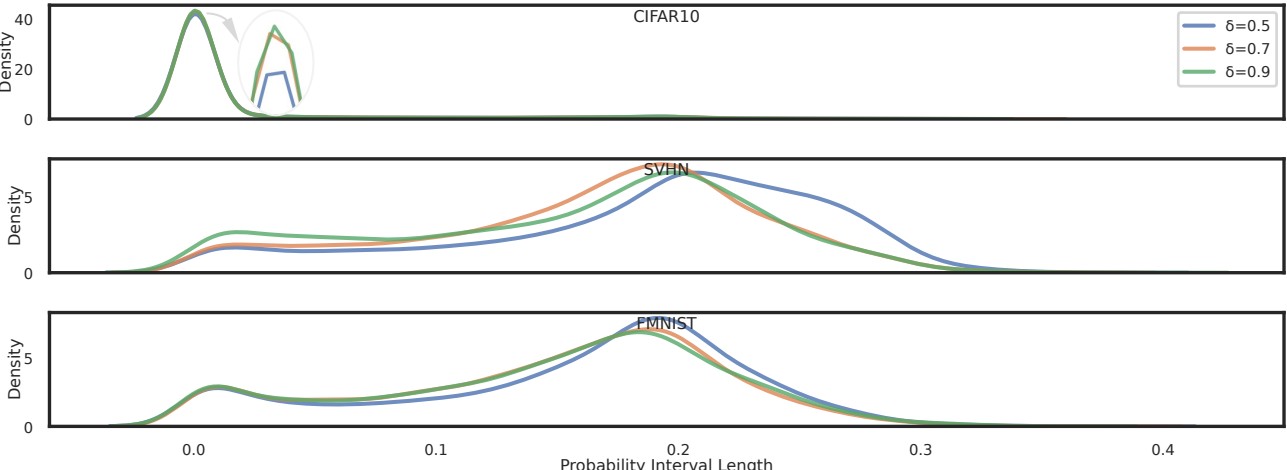

*Figure 11.* Kernel density plots of the averaged probability interval length (PIL) over classes on different datasets. The results empirically show that larger values of $\delta_G$ generally lead to larger box credal sets.

As shown in the Table 4, CreDRO's performance is stable across different choices of $\delta_G$. This is because $\delta_G$ reflects only one subjective belief from the model trainer about the worst-case scenario. In contrast, CreDRO training incorporates multiple subjective member beliefs about the potential distributional divergence (8), which reduces the significant impact of selecting a specific $\delta_G$. For example, an example showing how the selection of $\delta_G$ influences CreDRO training is provided in Table 12. A smaller value of $\delta_G$, which reflects a more pessimistic belief about potential distribution divergence, may produce larger box credal sets. To empirically examine this effect, we use the averaged probability interval length (PIL) as a measure of the credal set size (Löhr et al., 2025), defined as follows:

$$\text{PIL} = \frac{1}{C} \sum_{k=1}^{C} \overline{p}_k - \underline{p}_k. \tag{14}$$

*Table 12.* Example of the effect of $\delta_G$ selection on CreDRO training. Ensemble size $M = 5$.

| $\delta_G$ | Individuals' $\{\delta_i\}_{i=1}^{5}$ per Member | Number of Selected Top $\delta$ Samples per Batch |
|---|---|---|
| 0.5 | 0.50, 0.625, 0.75, 0.875, 1.00 | 64, 80, 96, 112, 128 |
| 0.7 | 0.70, 0.775, 0.85, 0.925, 1.00 | 89, 99, 108, 118, 128 |
| 0.9 | 0.90, 0.925, 0.95, 0.975, 1.00 | 115, 118, 121, 124, 128 |

A larger value of PIL approximately indicates a larger credal set. We do not adopt the entropy-based EU measure because it only indirectly reflects the credal set size, and credal sets with different sizes may yield similar (or identical) values (Hüllermeier et al., 2022). As shown in Figure 11, smaller values of $\delta_G$ generally lead to larger box credal sets, confirming that increasing pessimism about distribution divergence results in wider probability intervals.

# F. Ablation Study on Credal Set Constructions

As evaluated in Section 4.4, the box credal set $\mathcal{K}_B$ outperforms the convex hull $\mathcal{K}_C$ in OOD detection as measured by AUROC. In this appendix, we further report the False Positive Rate at 95% True Positive Rate (FPR@95TPR) and the mean epistemic uncertainty (EU) on CIFAR10 in-distribution (ID) test data in Table 13 and Table 14, respectively. These results quantitatively confirm that the superior performance of $\mathcal{K}_B$ is not attributable to overly conservative uncertainty estimation on ID samples. Furthermore, Table 15 reports the normalized EU gap between ID and OOD samples, demonstrating that $\mathcal{K}_B$ effectively widens this gap, which results in its improved OOD detection performance.

*Table 13.* FPR@95TPR (%) for OOD detection with CIFAR10 as the ID dataset ($\downarrow$), comparing $\mathcal{K}_B$ (box credal set) and $\mathcal{K}_C$ (convex hull credal set) under varying ensemble sizes ($M$). Results are averaged over 3 runs.

| | Method | SVHN | Places | CIFAR100 | FMNIST | ImageNet |
|---|---|---|---|---|---|---|
| $M = 5$ | $\mathcal{K}_B$ | **12.8**$_{\pm 1.4}$ | **30.4**$_{\pm 0.2}$ | **29.8**$_{\pm 0.3}$ | **16.7**$_{\pm 0.7}$ | **42.1**$_{\pm 1.9}$ |
| | $\mathcal{K}_C$ | 13.4$_{\pm 1.2}$ | 30.5$_{\pm 0.3}$ | 29.9$_{\pm 0.2}$ | 16.8$_{\pm 0.6}$ | 42.3$_{\pm 1.9}$ |
| $M = 10$ | $\mathcal{K}_B$ | **12.3**$_{\pm 0.9}$ | **29.6**$_{\pm 0.4}$ | **28.6**$_{\pm 0.3}$ | **16.4**$_{\pm 0.4}$ | **39.4**$_{\pm 1.1}$ |
| | $\mathcal{K}_C$ | 13.4$_{\pm 0.4}$ | 29.7$_{\pm 0.4}$ | 28.7$_{\pm 0.4}$ | 16.5$_{\pm 0.4}$ | 39.6$_{\pm 1.3}$ |
| $M = 15$ | $\mathcal{K}_B$ | **10.2**$_{\pm 2.1}$ | **29.7**$_{\pm 0.3}$ | **28.2**$_{\pm 0.2}$ | **16.7**$_{\pm 0.3}$ | **37.5**$_{\pm 1.0}$ |
| | $\mathcal{K}_C$ | 11.4$_{\pm 2.5}$ | 29.8$_{\pm 0.3}$ | 28.3$_{\pm 0.3}$ | 16.8$_{\pm 0.4}$ | 37.7$_{\pm 1.0}$ |
| $M = 20$ | $\mathcal{K}_B$ | **9.0**$_{\pm 0.4}$ | **29.7**$_{\pm 0.8}$ | 27.6$_{\pm 0.4}$ | **15.5**$_{\pm 1.0}$ | 37.2$_{\pm 0.7}$ |
| | $\mathcal{K}_C$ | 9.4$_{\pm 0.3}$ | 29.8$_{\pm 0.7}$ | 27.6$_{\pm 0.3}$ | 16.1$_{\pm 0.4}$ | 37.2$_{\pm 0.7}$ |

*Table 14.* Mean epistemic uncertainty estimates on CIFAR10 in-distribution (ID) test data for varying ensemble sizes ($M$), comparing $\mathcal{K}_B$ (box credal set) and $\mathcal{K}_C$ (convex hull credal set). $\mathcal{K}_B$ does not produce excessively large estimates on ID data.

| $M$ | $\mathcal{K}_B$ | $\mathcal{K}_C$ | Ratio ($\mathcal{K}_B/\mathcal{K}_C$) |
|---|---|---|---|
| 5 | 0.198$_{\pm 0.002}$ | 0.187$_{\pm 0.002}$ | 1.063 |
| 10 | 0.265$_{\pm 0.002}$ | 0.247$_{\pm 0.002}$ | 1.075 |
| 15 | 0.302$_{\pm 0.002}$ | 0.280$_{\pm 0.001}$ | 1.080 |
| 20 | 0.327$_{\pm 0.001}$ | 0.301$_{\pm 0.001}$ | 1.085 |

# G. Computational Cost of Epistemic Uncertainty Quantification

### G.1. Box credal set vs convex hull credal set

In this section, we analyze the computational cost of estimating epistemic uncertainty by computing the difference between the upper and lower entropy (Abellán et al., 2006) over two types of credal sets derived from softmax probability samples $\{\boldsymbol{p}_i\}_{i=1}^{M}$ for $C$ classes. Namely, we consider the box credal set $\mathcal{K}_B$, defined as

$$\mathcal{K}_B = \left\{ \boldsymbol{p} \mid p_k \in [\underline{p}_k, \overline{p}_k], \quad \sum_{k=1}^{C} p_k = 1 \right\},$$

*Table 15.* Normalized epistemic uncertainty gap ($\uparrow$), defined as $\left(\bar{\mathcal{U}}_{\text{OOD}} - \bar{\mathcal{U}}_{\text{ID}}\right)/\bar{\mathcal{U}}_{\text{OOD}}$, for $\mathcal{K}_B$ (box credal set) and $\mathcal{K}_C$ (convex hull credal set) across OOD detection data pairs, under varying ensemble sizes ($M$). Results are averaged over 3 runs.

|  |  | SVHN | Places | CIFAR100 | FMNIST | ImageNet |
|---|---|---|---|---|---|---|
| $M = 5$ | $\mathcal{K}_B$ | $\mathbf{0.882}_{\pm 0.070}$ | $\mathbf{0.849}_{\pm 0.001}$ | $\mathbf{0.845}_{\pm 0.003}$ | $\mathbf{0.867}_{\pm 0.012}$ | $\mathbf{0.841}_{\pm 0.006}$ |
|  | $\mathcal{K}_C$ | $0.876_{\pm 0.077}$ | $0.840_{\pm 0.001}$ | $0.837_{\pm 0.003}$ | $0.857_{\pm 0.014}$ | $0.833_{\pm 0.005}$ |
| $M = 10$ | $\mathcal{K}_B$ | $\mathbf{0.869}_{\pm 0.017}$ | $\mathbf{0.842}_{\pm 0.005}$ | $\mathbf{0.836}_{\pm 0.007}$ | $\mathbf{0.861}_{\pm 0.005}$ | $\mathbf{0.833}_{\pm 0.008}$ |
|  | $\mathcal{K}_C$ | $0.863_{\pm 0.018}$ | $0.833_{\pm 0.006}$ | $0.830_{\pm 0.007}$ | $0.852_{\pm 0.007}$ | $0.826_{\pm 0.008}$ |
| $M = 15$ | $\mathcal{K}_B$ | $\mathbf{0.862}_{\pm 0.020}$ | $\mathbf{0.836}_{\pm 0.003}$ | $\mathbf{0.830}_{\pm 0.004}$ | $\mathbf{0.855}_{\pm 0.006}$ | $\mathbf{0.827}_{\pm 0.004}$ |
|  | $\mathcal{K}_C$ | $0.855_{\pm 0.021}$ | $0.828_{\pm 0.003}$ | $0.824_{\pm 0.004}$ | $0.846_{\pm 0.008}$ | $0.820_{\pm 0.004}$ |
| $M = 20$ | $\mathcal{K}_B$ | $\mathbf{0.859}_{\pm 0.006}$ | $\mathbf{0.832}_{\pm 0.004}$ | $\mathbf{0.827}_{\pm 0.003}$ | $\mathbf{0.852}_{\pm 0.005}$ | $\mathbf{0.824}_{\pm 0.005}$ |
|  | $\mathcal{K}_C$ | $0.853_{\pm 0.005}$ | $0.825_{\pm 0.004}$ | $0.821_{\pm 0.004}$ | $0.845_{\pm 0.009}$ | $0.817_{\pm 0.005}$ |

where the upper and lower probability bound per class ($\bar{p}_k$ and $\underline{p}_k$) can be obtained from (9), and the convex hull credal set $\mathcal{K}_C$, defined by

$$\mathcal{K}_C = \left\{ \sum_{i=1}^{M} \pi_i \boldsymbol{p}_i \mid \pi_i \geq 0, \sum_{i=1}^{M} \pi_i = 1 \right\}.$$

**Box Credal Set Case.** Let $H(\boldsymbol{p}) := \sum_{i=1}^{C} -p_i \log p_i$ denote the entropy function over any valid probability vector. The optimizations for computing the *upper entropy* and *lower entropy* take the following forms:

$$
\begin{aligned}
\text{maximize}_{\boldsymbol{p}} \ H(\boldsymbol{p}) \quad &\text{s.t.} p_k \in [\underline{p}_k, \bar{p}_k] \quad \text{and} \ \sum_{k=1}^{C} p_k = 1 \\
\text{minimize}_{\boldsymbol{p}} \ H(\boldsymbol{p}) \quad &\text{s.t.} p_k \in [\underline{p}_k, \bar{p}_k] \quad \text{and} \ \sum_{k=1}^{C} p_k = 1
\end{aligned}
\tag{15}
$$

The optimization is performed over $\boldsymbol{p} \in \mathbb{R}^C$ with $C$ box constraints and one equality constraint. The objective $H(\boldsymbol{p})$ is strictly concave. In practice, we use `scipy.optimize.minimize` with bounds and equality constraints, initialized at the mean probability vector, as shown in Figure 12 (left). Regarding the computational complexity, each iteration requires $O(C)$ operations to compute the entropy and its gradient. The number of iterations $T_B$ depends on the solver and problem conditioning. Therefore, the total complexity per instance is $O(T_B C)$, which depends only on the number of classes $C$ and is independent of the number of samples $M$.

**Convex Hull Credal Set Case.** The convex hull credal set $\mathcal{K}_C$ corresponds to convex combinations of the original $M$ softmax samples. The optimizations for computing the *upper entropy* and *lower entropy* take the following forms:

$$
\begin{aligned}
\text{maximize}_{\boldsymbol{\pi} \in \Delta^{M-1}} \ H \left( \sum_{i=1}^{M} \pi_i \boldsymbol{p}_i \right) \\
\text{minimize}_{\boldsymbol{\pi} \in \Delta^{M-1}} \ H \left( \sum_{i=1}^{M} \pi_i \boldsymbol{p}_i \right)
\end{aligned}
\tag{16}
$$

where $\boldsymbol{\pi} \in \mathbb{R}^M$ lies in the $M$-simplex $\Delta^{M-1}$. The objective in (16) is concave in $\boldsymbol{\pi}$ and is optimized using `scipy.optimize.minimize` with $M$ box constraints $\pi_i \in [0, 1]$ and one equality constraint $\sum_{i=1}^{M} \pi_i = 1$. The full implementation is shown in Figure 12 (right). Regarding the computational complexity, each function evaluation involves computing the convex combination $\boldsymbol{p} = \sum_{i=1}^{M} \pi_i \boldsymbol{p}_i$ at a cost of $O(MC)$, plus an entropy evaluation costing $O(C)$. Each iteration thus costs $O(MC)$, with the total number of iterations $T_C$ depending on the solver and problem conditioning, resulting in a total complexity of $O(T_C MC)$ per instance. The complexity scales linearly with both the number of classes $C$ and the number of samples $M$.

**Summary of Computational Trade-offs.** From the analysis above, we observe that box credal set methods provide a more efficient alternative, with complexity that does not depend on $M$. This is particularly advantageous when the number of softmax samples $M$ is large, since optimizing the convex hull upper entropy becomes computationally expensive in that case. Note that computing the lower entropy remains inexpensive for both methods: for $\mathcal{K}_B$, the minimum occurs at an extreme point of the box, while for $\mathcal{K}_C$, it is attained at one of the original samples. A summary of the computational complexities for entropy optimization is given in Table 16.

*Table 16.* Summary of computational complexities for entropy optimization. $T_B, T_C$ denote the number of iterations of the numerical optimizer.

| | Quantity | Problem Size | Complexity per Instance |
|---|---|---|---|
| Box $\mathcal{K}_B$ | upper entropy | $C$ variables | $O(T_B \cdot C)$ |
| Box $\mathcal{K}_B$ | lower entropy | $C$ variables | $O(T_B \cdot C)$ |
| Convex hull $\mathcal{K}_C$ | upper entropy | $M$ variables | $O(T_C \cdot M \cdot C)$ |
| Convex hull $\mathcal{K}_C$ | lower entropy | trivial | $O(M \cdot C)$ |

```python
import numpy as np
from scipy.optimize import minimize
from scipy.stats import entropy
from tqdm import tqdm
def upper_entropy(probs: np.ndarray, base: float = 2) -> np.ndarray:
    """Compute the upper entropy of a box credal set.
    Args:
        probs: numpy.ndarray of shape (n_instances, n_samples, n_classes)
        base: float, default=2
    Returns:
        ue: numpy.ndarray of shape (n_instances,)
    """
    def fun(x: np.ndarray) -> np.ndarray:
        return -entropy(x, base=base)

    x0 = probs.mean(axis=1)
    constraints = {"type": "eq", "fun": lambda x: np.sum(x) - 1}
    ue = np.empty(probs.shape[0])
    for i in tqdm(range(probs.shape[0])):
        bounds = list(zip(np.min(probs[i], axis=0), np.max(probs[i], axis=0), strict=False))
        res = minimize(fun=fun, x0=x0[i], bounds=bounds, constraints=constraints)
        ue[i] = -res.fun
    return ue

def lower_entropy(probs: np.ndarray, base: float = 2) -> np.ndarray:
    """Compute the lower entropy of a box credal set.
    Args:
        probs: numpy.ndarray of shape (n_instances, n_samples, n_classes)
        base: float, default=2
    Returns:
        le: numpy.ndarray of shape (n_instances,)
    """
    def fun(x: np.ndarray) -> np.ndarray:
        return entropy(x, base=base)
    MINIMIZE_EPS = 0.001
    x0 = probs.mean(axis=1)
    # If the initial solution is uniform, slightly perturb it, as minimize will fail otherwise
    uniform_idxs = np.all(np.isclose(x0, 1 / probs.shape[2]), axis=1)
    x0[uniform_idxs, 0] += MINIMIZE_EPS
    x0[uniform_idxs, 1] -= MINIMIZE_EPS

    constraints = {"type": "eq", "fun": lambda x: np.sum(x) - 1}
    le = np.empty(probs.shape[0])
    for i in tqdm(range(probs.shape[0])):
        bounds = list(zip(np.min(probs[i], axis=0), np.max(probs[i], axis=0), strict=False))
        res = minimize(fun=fun, x0=x0[i], bounds=bounds, constraints=constraints)
        le[i] = res.fun
    return le
```

```python
import numpy as np
from scipy.optimize import minimize
from scipy.stats import entropy
from tqdm import tqdm
def upper_entropy_convex_hull(probs: np.ndarray, base: float = 2) -> np.ndarray:
    """Compute the upper entropy of a convex hull credal set.
    Args:
        probs: numpy.ndarray of shape (n_instances, n_samples, n_classes)
        base: float, default=2
    Returns:
        ue: numpy.ndarray of shape (n_instances,)
    """
    def fun(w: np.ndarray, extrema: np.ndarray) -> np.ndarray:
        prob = w @ extrema
        return -entropy(prob, base=base)

    w0 = np.ones(probs.shape[1]) / probs.shape[1]
    constraints = {"type": "eq", "fun": lambda x: np.sum(x) - 1}
    bounds = [(0, 1)] * probs.shape[1]
    ue = np.empty(probs.shape[0])
    for i in tqdm(range(probs.shape[0])):
        res = minimize(fun=fun, args=probs[i], x0=w0, bounds=bounds, constraints=constraints)
        ue[i] = -res.fun
    return ue

def lower_entropy_convex_hull(probs: np.ndarray, base: float = 2) -> np.ndarray:
    """Compute the lower entropy of a convex hull credal set.
    Args:
        probs: numpy.ndarray of shape (n_instances, n_samples, n_classes)
        base: float, default=2
    Returns:
        le: numpy.ndarray of shape (n_instances,)
    """
    def fun(w: np.ndarray, extrema: np.ndarray) -> np.ndarray:
        prob = w @ extrema
        return entropy(prob, base=base)

    w0 = np.ones(probs.shape[1]) / probs.shape[1]
    constraints = {"type": "eq", "fun": lambda x: np.sum(x) - 1}
    bounds = [(0, 1)] * probs.shape[1]
    le = np.empty(probs.shape[0])
    for i in tqdm(range(probs.shape[0])):
        res = minimize(fun=fun, args=probs[i], x0=w0, bounds=bounds, constraints=constraints)
        le[i] = res.fun
    return le
```

*Figure 12.* Code implementation for computing the entropy difference of the box credal set (left) and the convex hull credal set (right), following Löhr et al. (2025).

**G.2. Empirical analysis of effect of interval tightness on optimization runtime for $\mathcal{K}_B$**

In our implementation, the upper and lower entropy over $\mathcal{K}_B$ are computed using `scipy.optimize.minimize` with box constraints and a simplex equality constraint, as discussed in Appendix G.1. The runtime depends non-monotonically on the width of the probability intervals. When the intervals are moderately tight, the feasible region is small, and the initialization at the mean probability vector is typically close to the optimum, which could lead to fast convergence in a few iterations. For wide intervals, the optimizer explores a larger feasible region and requires more iterations. Conversely, when the intervals become extremely tight, the feasible polytope may become nearly degenerate, with many active bound constraints interacting with the normalization constraint. This can lead to ill-conditioning and slower convergence for SLSQP-style solvers. Empirically, Table 17 reports the averaged probability interval length (PIL) on the CIFAR10 test data of several credal classifiers and their UQ runtime. CreDRO achieves the fastest regime, while too-tight probability intervals of CreDEs lead to a higher UQ computation cost.

*Table 17.* Averaged probability interval length (PIL $= \frac{1}{10} \sum_{k=1}^{10} \bar{p}_k - \underline{p}_k$) and UQ runtime (seconds) on CIFAR10 test instances.

|              | PIL (Run #1)        | PIL (Run #2)        | PIL (Run #3)        | Averaged UQ Runtime |
|--------------|---------------------|---------------------|---------------------|---------------------|
| CreDRO       | $0.0179_{\pm 0.0498}$ | $0.0180_{\pm 0.0501}$ | $0.0182_{\pm 0.0506}$ | $116.37_{\pm 0.30}$   |
| CreWra       | $0.0216_{\pm 0.0557}$ | $0.0207_{\pm 0.0539}$ | $0.0212_{\pm 0.0551}$ | $123.83_{\pm 0.24}$   |
| CreRL$_{1.0}$ | $0.0271_{\pm 0.0636}$ | $0.0274_{\pm 0.0643}$ | $0.0263_{\pm 0.0633}$ | $133.31_{\pm 1.04}$   |
| CreDE        | $0.0009_{\pm 0.0025}$ | $0.0009_{\pm 0.0025}$ | $0.0009_{\pm 0.0025}$ | $165.20_{\pm 0.96}$   |

# H. Possible Extension to Regression

In this section, we illustrate a possible way of formulating credal predictions from our CreDRO in regression settings.

Note that the training framework is in principle applicable to regression, as the training objective $\mathcal{L}(\cdot, \cdot)$, outlined in Algorithm 1, could be any suitable loss, provided it enables neural networks to produce probabilistic predictions. E.g., the Gaussian negative log-likelihood loss (Lakshminarayanan et al., 2017) for regression tasks.

In a regression setting with a continuous target space $\mathbb{R}$, our analysis considers a set of Gaussian predictions produced by an ensemble, denoted $\{\mathcal{N}(\mu_i, \sigma_i^2)\}_{i=1}^M$. Gaussian distributions are chosen because mean-variance neural network estimators are relatively straightforward to learn from data and widely applied in practice (Lakshminarayanan et al., 2017; Sluijterman et al., 2024; Lehmann et al., 2024). In this setting, the final single precise prediction is approximated by a single Gaussian distribution, denoted as $\mathcal{N}(\mu_G, \sigma_G^2)$, constructed by matching the mean and variance of the corresponding mixture distribution (Lakshminarayanan et al., 2017). The resulting mean is given by $\mu_G = M^{-1} \sum_{i=1}^M \mu_i$, and the variance is computed as

$$\sigma_G^2 = \frac{1}{M} \sum_{i=1}^M (\sigma_i^2 + \mu_i^2) - \mu_G^2. \tag{17}$$

Under the context of a continuous target space, extracting probability intervals over discrete classes as in (9) is not applicable. Consequently, CreDRO could map the ensemble's Gaussian predictions, $\{\mathcal{N}(\mu_i, \sigma_i^2)\}_{i=1}^M$, to a finitely generated credal set (Augustin et al., 2014; Chau et al., 2025b). This credal set $\mathcal{K}_F$ is defined as

$$\mathcal{K}_F = \left\{ \sum_{i=1}^M \pi_i \mathcal{N}(\mu_i, \sigma_i^2) \,\middle|\, \pi_i \geq 0, \sum_{i=1}^M \pi_i = 1 \right\}. \tag{18}$$

Hence, $\mathcal{K}_F$ represents a set of mixtures of Gaussians. When $\pi_i = M^{-1}$ for all $i$, $\mathcal{K}_F$ reduces to the final prediction $\mathcal{N}(\mu_G, \sigma_G^2)$ in classical ensembles (Lakshminarayanan et al., 2017), given by (17).

From the above analysis, the way of applying credal predictions of our CreDRO in regression demonstrates promising potential. However, compared to classification, credal methods for uncertainty quantification in regression remain less mature, with fewer studies on uncertainty measures and their evaluation protocols. Therefore, more rigorous theoretical analysis and more comprehensive empirical validation in regression are left for future work.

