# OpenReview forum: "Learning Credal Ensembles via Distributionally Robust Optimization"
_ICML.cc/2026/Conference — ICML 2026 spotlight_

### Official Review · Reviewer_DKpD · 2026-03-11

**Soundness:** 3
**Presentation:** 3
**Significance:** 4
**Originality:** 4
**Overall Recommendation:** 6
**Confidence:** 4

**Summary:**

This paper introduces CreDRO, an ensemble framework designed to quantify epistemic uncertainty by moving beyond standard methods that define EU solely as disagreement caused by random training initializations. Instead, I appreciate how the authors formulate EU as the disagreement among models trained under varying degrees of simulated train-test distribution shifts using distributionally robust optimization DRO. To achieve this efficiently, the method employs a batch wise heuristic where only the top portion of samples with the highest loss are used for backpropagation, simulating potential domain shifts. A global hyperparameter is used to uniformly interpolate these shift degrees across the individual ensemble members. During inference, the ensemble's independent softmax probabilities are converted into class wise probability intervals to form a box credal set. The framework's epistemic uncertainty is then quantified using the difference between upper and lower Shannon entropy. The validation across multiple OOD detection benchmarks and a selective classification task using the Camelyon17 medical dataset is a strong addition, particularly since robust UQ is so vital in clinical AI screening.

**Compliance With Llm Reviewing Policy:**

Affirmed.

**Final Justification:**

My initial evaluation praised the practical utility of CreDRO for safety-critical diagnostic tasks and its elegant approach to distributionally robust optimization (DRO). However, I held significant reservations regarding the lack of mathematical rigor bridging the minimax objective to the discrete batch-wise sample dropping, as well as the heuristic assumption that top-loss samples represent domain shifts rather than simple label noise.

The authors provided an effective, mathematically rigorous, and data-driven rebuttal that fully resolved my concerns:
- Theoretical Soundness: The authors provided the formal mathematical derivation linking the conditional value at risk (CVaR) to the top-alpha loss samples. This successfully proved that their batch-wise sample dropping is a direct empirical approximation of the full-batch CVaR objective, entirely removing the "heuristic" label from their methodology.
- Robustness to Label Noise: The authors ran new experiments injecting 10% and 20% symmetric label noise into CIFAR10 and introduced a random-reweighting baseline (CreRAM). The results decisively proved that CreDRO's performance gains are driven by semantically guided minority-group selection rather than arbitrarily fitting noisy outliers.
- Hyperparameter Robustness: New ablation studies demonstrated that the framework is highly robust to the distributional assignment of the relaxation parameter alpha, validating the authors' uniform prior design choice.
By closing the theoretical loop and providing robust empirical validation against label noise, the authors have significantly elevated the quality of this manuscript. This is a highly accessible, rigorously validated framework for epistemic uncertainty quantification. I have raised my score and strongly advocate for its acceptance.

**Key Questions For Authors:**

1. How does the top loss sample selection mechanism differentiate between hard to learn minority group samples and inherently noisy or mislabeled instances?

2. The uniform interpolation of the relaxation parameter between the global bound and 1 is presented as a design choice. Have you experimented with other distributional assignments for these parameters across the ensemble?

3. Since the box credal set inherently ignores the dependency structure between classes compared to the convex hull, does this loss of structural information noticeably impact calibration in datasets with highly correlated class structures?

**Limitations:**

While the authors acknowledge limitations regarding the derivation of single predictive probabilities from credal sets and the current restriction to classification tasks, the manuscript would benefit from a dedicated discussion outlining the framework's potential vulnerability to label noise. This is particularly important given that the DRO training objective heavily prioritizes high loss samples during backpropagation.

**Strengths And Weaknesses:**

Soundness
- Strengths: Reframing epistemic uncertainty to explicitly account for train-test distribution shifts is a well motivated and practically useful departure from standard deep ensembles. I find the approximation of group DRO via top loss sample selection to be a computationally viable strategy that avoids the inefficiency of directly estimating adversarial weights.
- Weaknesses: My main concern is the assumption that high loss training instances cleanly correspond to minority groups simulating test-time domain shifts; this is ultimately a heuristic. In real world datasets with significant label noise, this hard mining strategy risks forcing ensemble members to fit noisy outliers rather than capturing meaningful structural shifts.

Presentation
- Strengths: The manuscript is well structured and clearly delineates the differences between CreDRO and competing credal methods. I appreciate that it specifically highlights a practical advantage over the recent CreDE baseline, noting that CreDRO requires no architectural modifications to the output layers. The ablation studies on the global hyperparameter and credal set structures, box vs convex hull, are thorough.
- Weaknesses: The transition from the formal minimax objective of adversarially reweighted learning to the discrete batch wise sample dropping could be mathematically bridged with more rigor.

Significance
- Strengths: By offering a method that utilizes standard neural network architectures and standard cross entropy loss without requiring one hot label constraints, CreDRO is highly accessible. The verified performance gains on realistic diagnostic tasks like histopathology image classification indicate strong utility for safety-critical applications, which aligns well with my own work in the field.

Originality
- Strengths: The mechanism of generating ensemble diversity by uniformly interpolating the DRO relaxation parameter across individual members is an innovative and elegant approach to capturing varying degrees of distributional divergence.

---

> ### Author Rebuttal · Authors · 2026-03-30
>
> Dear Reviewer,
>
> We sincerely appreciate your valuable feedback and your positive assessment of our work. Below, we clarify the points you raised.
>
> **W2:** Following your useful comments, we provide the mathematical bridge as follows. We instantiate $\mathbb{W}$ in Eq.(4) as a conditional value at risk (CVaR) uncertainty set at level $\delta$ [1]:
>
> $\mathbb{W}=\left\lbrace\boldsymbol{w}\geq\boldsymbol{0} \mid \sum_{n=1}^{N} w_n = N, w_n \leq \delta^{-1},\forall n \right\rbrace.$
>
> The inner maximization in Eq.(4) then admits a closed-form solution-the optimal adversary assigns weight $\delta^{-1}$ to the top-$\lfloor \delta N \rfloor$ loss samples and zero otherwise:
>
> $\max_{\boldsymbol{w} \in \mathbb{W}}\frac{1}{N}\sum_{n=1}^{N}w_n {\mathcal{L}}_n(\cdot,\cdot)=\frac{1}{\delta N}\sum\_{n \in \mathcal{S}\_{\delta}} \mathcal{L}_n(\cdot,\cdot),$
>
> where $\mathcal{S}_\delta$ is the top-$\delta$ highest-loss indices. This follows directly from a linear program over the simplex, where the optimum concentrates mass on the worst-$\delta$ fraction. Batch-wise top-$\delta$ selection is therefore not a heuristic but a direct empirical approximation of this full-batch CVaR objective. We will incorporate this into the revision.
>
> **W1&Q1&Limit:** Thanks for your insightful comments. Upweighting high-loss samples to emphasize the tail of the empirical distribution is a well-established instantiation of the DRO framework [1, 2, 3], and we acknowledge the structural vulnerability you identify, particularly when no additional information about the training data is available: mislabeled and minority-group samples could both exhibit elevated losses.
>
> We argue, however, that these two sources of high loss are distinct in practice. Noisy labels tend to produce erratic, inconsistent loss signals across training, whereas minority-group samples drive systematic and structurally consistent disagreement due to underrepresentation, making them more stably selected under the top-$\delta$ criterion over the course of training. Mislabeled instances are unlikely to dominate the top-$\delta$ selection consistently across training epochs.
>
> To *empirically* validate this, we inject symmetric label noise at 10% and 20% into CIFAR10 training data while keeping OOD test sets clean. To isolate the contribution of top-loss selection from mere sample reweighting, we further introduce *CreRAM*, which randomly upweights a $\delta$ fraction of batch samples without loss-guided selection.
>
> In Tab1&2 (*https://anonymous.4open.science/r/icml26rebuttal-433F/rebuttal.pdf*), CreDRO consistently outperforms both CreRAM and non-DRO baselines under all noise levels, showing that performance gains stem from semantically guided selection rather than arbitrary reweighting.
>
> We note that while label noise robustness falls outside the primary scope of this work, we agree that a dedicated discussion on label noise robustness is valuable, and we will incorporate this into the revision. We also identify label-noise-aware CreDRO extension as an important direction for future work.
>
> **Q2:** The uniform interpolation is indeed a design choice, reflecting a *uniform prior* over $[\delta_G, 1]$: in the absence of domain-specific evidence favoring any particular $\delta$, all values are treated as equally plausible.
>
> Inspired by your question, we conduct an ablation study comparing uniform, exponential-like (concentrated near $\delta_G$) and logarithmic-like (concentrated near 1) strategies. In Tab.3 (*https://anonymous.4open.science/r/icml26rebuttal-433F/rebuttal.pdf*), OOD detection performance is nearly identical across all three, supporting CreDRO's robustness to this design choice.
>
> **Q3:** Thanks for raising this point. We believe that this loss of structural information does not noticeably impact calibration in practice. The averaged probability vector, used as the point prediction for accuracy and calibration evaluation, is identical under both representations, as both credal sets are constructed from the same collection of ensemble probability vectors. Consequently, their calibration and prediction performance are equivalent.
>
> Further, on in-distribution (ID) data, ensemble members tend to produce concentrated and similar probability vectors, meaning the box credal set does not yield excessively large or poorly structured credal sets. To verify this, we report the averaged EU estimates of $K_B$ (box) and $K_C$ (convex hull) and their ratio ($K_B$/$K_C$) in Tab4 (*https://anonymous.4open.science/r/icml26rebuttal-433F/rebuttal.pdf*), showing that $K_B$ does not produce excessively large estimates on ID data.
>
> *We hope these responses adequately address your concerns.*
>
> [1] Daniel Levy et al. Large-Scale Methods for Distributionally Robust Optimization. NeurIPS2020.
>
> [2] Anurag Singh et al. Domain Generalisation via Imprecise Learning. ICML2024.
>
> [3] John C. Duchi, et al. Learning Models with Uniform Performance via Distributionally Robust Optimization.2021.

---

> > ### Author Rebuttal · Reviewer_DKpD · 2026-04-02
> >
> > To be honest, I am highly impressed. It is rare for a rebuttal to not only provide new empirical data but also completely close a theoretical gap identified in the review. You have definitively elevated this paper.
> > - Regarding the mathematical rigor (W2): Providing the explicit formulation linking the conditional value at risk (CVaR) uncertainty set to the top-alpha loss samples was exactly what this manuscript needed. By demonstrating that the batch-wise top-K selection is a direct empirical approximation of the full-batch CVaR objective rather than a mere heuristic, you have completely solidified the theoretical soundness of CreDRO.
> > - Regarding label noise vs. domain shift (W1/Q1): Your argument that noisy labels produce erratic loss signals while minority-group samples produce consistent signals is logically sound. More importantly, you backed it up. The injection of 10% and 20% symmetric label noise into CIFAR10, coupled with the ablation against the non-loss-guided CreRAM, definitively proves that CreDRO's gains stem from semantically guided selection. Any lingering concerns I had regarding the model's vulnerability to noisy outliers is completely alleviated.
> > - Regarding the alpha distribution and credal sets (Q2/Q3): Thank you for running the additional ablation on the uniform vs. exponential/logarithmic alpha distributions. Seeing that the OOD performance remains robust regardless of the prior validates your design choice. Furthermore, clarifying that the point prediction is identical under both representations adequately resolves my concern regarding calibration and structural loss.
> > You have addressed every single weakness with mathematical rigor and hard data. I expect the CVaR derivation and the label noise experiments to be prominently featured in the final manuscript.

---

> > > ### Author Response · Authors · 2026-04-03
> > >
> > > Dear Reviewer,
> > >
> > > Thank you again for your recognition and support for our work. We are glad that our responses have addressed your concerns. Your constructive comments have been valuable in improving the quality of this work, and we will carefully incorporate them into the revision.
> > >
> > > We sincerely appreciate your review effort and constructive comments again.
> > >
> > > With all the best regards,
> > >
> > > The Authors

---

### Official Review · Reviewer_E2oi · 2026-03-12

**Soundness:** 2
**Presentation:** 3
**Significance:** 2
**Originality:** 3
**Overall Recommendation:** 4
**Confidence:** 4

**Summary:**

The paper proposes the CreDRO framework to quantify epistemic uncertainty (EU) in deep neural networks. The authors argue that currently the most advanced methods, such as standard deep ensembles, define EU as the difference caused by random initialization, which only captures optimization noise, rather than knowledge gaps. To solve the problem, CreDRO uses Distributionally Robust Optimization (DRO) to an ensemble. CreDRO applies hyperparameter δ_i to expose each member to varying degrees of simulated distribution shifts. During inference, the ensemble output is converted into box-based credal sets to represent uncertainty.

**Compliance With Llm Reviewing Policy:**

Affirmed.

**Final Justification:**

This paper proposes CreDRO framework, which uses Distributionally Robust Optimization (DRO) to train an ensemble of models with different robustness characteristics. By varying the DRO hyperparameter, it effectively simulates potential train-test distribution shifts to quantify epistemic uncertainty (EU) via box credal sets.

Initially, I expressed concerns regarding the theoretical justification of using loss-based reweighting to simulate semantic shifts and the potential over-conservatism of the box-based credal approximation. However, the authors' comprehensive rebuttal has effectively addressed these points.

Overall, The rebuttal provided solid empirical evidence and necessary theoretical clarifications. The proposed method is practical and shows clear advantages over SOTA baselines in quantifying informative EU. I recommend a Weak Accept.

**Key Questions For Authors:**

Question
1. Can the authors provide a formal theoretical explanation that changing δ_i does indeed induce differences in the robustness of the model to semantic distribution shifts, rather than just artificial sample reweighting noise?
2. The paper claims that CreDRO has robustness in selecting δ_G, but ablation experiments only tested values between 0.5 and 0.9. What happens when δ_G is set to 0.5 or below? Will EU signals become meaningless? If so, how does this reconcile with the claim of robustness?
3. Although ablation experiments showed that K_B outperformed K_c in OOD detection of AUROC, the authors only attributed it to "widening the EU gap between ID/OOD". Please provide quantitative analysis to demonstrate that this performance improvement is not due to overly conservative estimation of ID samples. Specifically, report the FPR@95TPR (False Positive Rate at 95% True Positive Rate) and the Expected Calibration Error (ECE) for the ID set, to ensure that the box-based credal approximation has not compromised the predictive reliability of the model.

**Limitations:**

yes

**Strengths And Weaknesses:**

Strength:
1.    The paper’s motivation is intuitive. It is reasonable to criticize that random initialization is a poor substitute indicator of cognitive uncertainty and to identify the gaps in the literature.
2.	Different form CreDE, CreDRO does not require modification of the model layer structure and is compatible with standard neural network architectures. The training process relies on the simple straightforward sample reweighting, which is easy to understand and friendly to the parctitioner.
3.	The study contains a multi-dimensional assessment. It includes both standard OOD detection benchmarks and robustness tests on corrupted datasets (CIFAR10-C/CIFAR100-C) . Furthermore, multiple ablation experiments were conducted, including ensemble scale, hyperparameters, and credal set construction methods, which not only verified the optimal performance of the method, but also provided a clearer breakdown of the mechanism of the core modules.


Weakness:

1. The entire training process relies on an unverified assumption: the linear interval of δ_i can simulate a series of distribution shifts. If varying δ_i actually cannot induce model robustness to semantic shifts (e.g., weather conditions in autonomous driving or scanner variability in medical imaging) , the induced divergence is just another form of artificial noise, which is not more significant than the random initialization.
2. The authors only conducted ablation experiments with δ_G>=0.5 , and there is no rigorous sensitivity analysis to demonstrate whether different interpolation strategies,such as exponential or logarithmic spacing, are better. There is also no experimental analysis whether there is robustness for δ_G values below 0.5.
3. Box-based cedal set is a calculation shortcut which introduces unknown bias. By taking the minimum/maximum softmax probability between integrated members, the authors actually amplify the uncertainty boundary, which may lead to overly conservative predictions and false positives in OOD detection. Also, the authors’ experiment only proves that the box-based credal focuses exclusively on OOD performance gains whithout addressing whether this comes at the cost of ID prediction performance.

---

> ### Author Rebuttal · Authors · 2026-03-31
>
> Dear Reviewer,
>
> We are grateful for your valuable feedback and for acknowledging our work. Below, we address your concerns.
>
> **W1&Q1:** Following your constructive comments, we provide a *theoretical* clarification and *additional empirical* evidence.
>
> *Theoretically*, in our implementation, we instantiate the uncertainty set $\mathbb{W}$ in Eq.4 in the paper as a conditional value at risk (CVaR) at level $\delta$:
>
> $\mathbb{W}=\left\lbrace\boldsymbol{w}\geq\boldsymbol{0} \mid \sum_{n=1}^{N} w_n = N, w_n \leq \delta^{-1},\forall n \right\rbrace.$
>
> A model trained with a smaller $\delta$ corresponds to a strictly more conservative (worst-case) uncertainty set, i.e., $\delta_1 < \delta_2 \implies \mathbb{W}(\delta_1) \supseteq \mathbb{W}(\delta_2)$. By assigning each member a distinct $\delta_i$, CreDRO constructs models with formally distinct robustness profiles, rather than merely reweighting samples arbitrarily.
>
> Prior work in DRO, e.g. [1, 2, 3], has shown that worst-case reweighting tends to emphasize underrepresented or hard subpopulations, which are often associated with distributional shifts. While semantic shifts are not explicitly labeled during training, they frequently manifest as systematically higher loss regions under empirical risk minimization, making them more likely to be captured by the top-$\delta$ selection. We note that this connection is not an assumption unique to our method, but a standard interpretation in the field.
>
> *Empirically*, we verify that the gains are not due to arbitrary reweighting. Beyond real-world medical imaging experiments involving scanner variability (Sec. 4.6), where CreDRO outperforms baselines under practical semantic shifts, we further evaluate robustness under label noise: Inject 10% and 20% symmetric noise into CIFAR10 training data and keep OOD benchmarks clean.
>
> To isolate the effect of structured loss-guided selection, we introduce CreRAM, which randomly upweights a $\delta$ fraction of samples without loss guidance. In Tab.1&2 (*https://anonymous.4open.science/r/rebuttal26icml-456E/RebuttalICML.pdf*), CreDRO consistently outperforms both CreRAM and non-DRO baselines, indicating that the improvements arise from structured, loss-guided selection rather than arbitrary reweighting.
>
> **W2&Q2:** *Regarding uniform interpolatation:* In the absence of prior knowledge favoring any particular $\delta$, uniform interpolation is the natural assumption-free choice, ensuring the $M$ ensemble members are evenly spread across $[\delta_G, 1]$, with no region over- or under-represented.
>
> Inspired by your comment, we conduct a study comparing uniform, exponential-like (concentrated near $\delta_G$), and logarithmic-like (concentrated near 1) strategies. In Tab.3 (*https://anonymous.4open.science/r/rebuttal26icml-456E/RebuttalICML.pdf*), OOD detection performance is nearly identical across all three, showing CreDRO's robustness to this design choice.
>
> *Regarding $\delta_G$:* Setting $\delta_G < 0.5$ forces the lowest-indexed ensemble member to train on an excessively small subset of high-loss samples (e.g., $\delta_G = 0.3$ yields 38 samples per batch of 128), resulting in large and unstable gradient updates that can disrupt training, especially in the early stages. This issue is primarily numerical rather than a fundamental limitation of the EU signal. Therefore, our robustness claim is restricted to $\delta_G \in [0.5, 1)$, as supported by the ablation results. In our extensive evaluation, we adopt $\delta_G = 0.5$ as the default setting, under which CreDRO consistently outperforms the baselines, demonstrating that CreDRO's informative EU estimation.
>
> **W3&Q3:** Both credal sets are constructed from the same ensemble probability vectors, and the averaged probability vector, used as the point prediction for accuracy and calibration evaluation, is identical under both representations. Thus, these two yield equivalent ECE and accuracy (as shown in Tab.2 in the paper) on ID data.
>
> Following your suggestion, we report additional results in the anonymous link (*https://anonymous.4open.science/r/rebuttal26icml-456E/RebuttalICML.pdf*): FPR@95TPR (Tab.4) confirms the consistently superior OOD detection performance of $\mathcal{K}_B$ over $\mathcal{K}_C$; averaged EU estimates on ID data (Tab.5) confirm that $\mathcal{K}_B$ does not produce excessively large ID uncertainty; and the normalized EU gap (Tab.6) supports that the performance gains of $\mathcal{K}_B$ stem from a wider EU separation between ID and OOD samples, without generating overconservative ID estimation.
>
> *We hope these responses adequately address your concerns.*
>
> [1] Daniel Levy et al. Large-Scale Methods for Distributionally Robust Optimization. NeurIPS 2020.
>
> [2] Zeyi Huang et al. The Two Dimensions of Worst-Case Training and Their Integrated Effect for Out-of-Domain Generalization. CVPR 2022.
>
> [3] John C. Duchi, et al. Learning Models with Uniform Performance via Distributionally Robust Optimization. 2021.

---

> > ### Author Rebuttal · Reviewer_E2oi · 2026-04-02
> >
> > I would like to thank the authors for their comprehensive and highly constructive rebuttal. The additional experiments and theoretical clarification completely resolved my initial concerns.
> >
> > (1)The formal connection with CVaR, most importantly, the new CreRAM baseline convincingly demonstrates that performance gains come from structured, loss oriented selection rather than arbitrary reweighting noise.
> >
> > (2)The additional experiments on testing the exponential and logarithmic interpolation strategies were very helpful and confirmed the robustness of the method.
> >
> > (3)Newly provided FPR@95TPR The results and ID EU estimation effectively alleviated my concern about the overly conservative estimation of KB on the ID sample.
> >
> > Given the high quality of this rebuttal and the solid additional empirical evidence provided, I am satisfied with the response.

---

> > > ### Author Response · Authors · 2026-04-02
> > >
> > > Dear reviewer,
> > >
> > > Thank you again for your thoughtful engagement with our submission and for your positive recognition of our rebuttal. We truly appreciate that you feel our responses have fully addressed your concerns. Should you feel that our work now merits a higher score, we would be most grateful. Thank you again for your time, understanding, and support!
> > >
> > > Best regards,
> > >
> > > The authors

---

### Official Review · Reviewer_dHh4 · 2026-03-13

**Soundness:** 2
**Presentation:** 3
**Significance:** 3
**Originality:** 2
**Overall Recommendation:** 4
**Confidence:** 3

**Summary:**

The authors propose a method called CreDRO for quantifying epistemic uncertainty. Based on Distributionally Robust Optimization, this method  learns an ensemble of plausible models by varying a weight hyperparameter to simulate different levels of train–test
distribution shift during training. Then, author extensively benchmark their method on multiple settings in order to show its relevance (for out-of-distribution detection and selective classification)

**Compliance With Llm Reviewing Policy:**

Affirmed.

**Final Justification:**

The reason of my score are:
1\ All my concerned have been addressed during the rebuttal.
2\Furthermore, the other reviewers are positive about the paper.

**Key Questions For Authors:**

Q1. What happens when the number of models M increases?

Q2. Can you explain how you approach OOD using EN-DRO? More generally, if I am not mistaken, the way in which OOD is carried out using credal set is not explained.

Q3. The results are calculated as an average over 3 runs, which seems rather few. Why not carry out more runs?

Q4. In Table 4, the impact of delta appears to be very small. What would you recommend for adjusting this parameter?

Minor question: the results for EN-DRO are very close to those of CreDRO, but without the need to calculate sets of credaux. Is this set of credaux for OOD really useful?

**Limitations:**

Yes, the limitations are discussed in the conclusion.

**Strengths And Weaknesses:**

Strengths:

- This article is pleasant to read and easy to follow.

- The subject of interest is important

- The combination of credal set and distributionnaly robust optimization seems cleaver.

---
Weaknesses:

- The main weakness of the paper is that the various assertions and choices (for example, regarding the delta) are never supported by theoretical results or evidence.

- If I am correct, the code is not provided in the supplementary material.

---

> ### Author Rebuttal · Authors · 2026-03-30
>
> Dear Reviewer,
>
> We sincerely thank you for your review effort and detailed feedback. We respond to your comments below.
>
> **W1:** Thanks for your inspiring comments. The choice of $\delta$ is grounded in both theory and empirical evidence.
>
> *Theoretically*, $\delta$ is not a free hyperparameter—it is the conditional value at risk (CVaR) level in a principled DRO framework, where a smaller $\delta$ directly corresponds to a stricter worst-case objective—a well-defined quantity with established theoretical [1, 2] and applied [3] support.
>
> Concretely, our uncertainty set $\mathbb{W}$ is directly adapted from the CVaR formulation in [1], with the probabilistic weights replaced by normalized weights to fit our setting:
>
> $\mathbb{W}=\left\lbrace\boldsymbol{w}\geq\boldsymbol{0} \middle| \sum_{n=1}^{N} w_n = N, w_n \leq \delta^{-1},\forall n \right\rbrace.$
>
> The inner maximization in Eq.(4) of our paper then admits a closed-form solution—the optimal adversary assigns weight $\delta^{-1}$ to the top-$\lfloor \delta N \rfloor$ highest-loss samples and zero otherwise:
>
> $\max_{\boldsymbol{w} \in \mathbb{W}}\frac{1}{N}\sum_{n=1}^{N}w_n {\mathcal{L}}_n(\cdot,\cdot)=\frac{1}{\delta N}\sum\_{n \in \mathcal{S}\_{\delta}} \mathcal{L}_n(\cdot,\cdot),$
>
> where $\mathcal{S}\_\delta$ is the top-$\delta$ highest-loss sample set. A model trained with a smaller $\delta$ corresponds to a strictly more conservative (worst-case) uncertainty set, i.e., $\delta_1 < \delta_2 \implies \mathbb{W}(\delta_1) \supseteq \mathbb{W}(\delta_2)$. Thus, $\delta$ has a clear interpretation rather than being an arbitrary choice. We will incorporate these clarifications into the revision.
>
> *Empirically*, we set $\delta_G = 0.5$ as the default and evaluate *CreDRO* across multiple benchmarks—standard OOD detection, corrupted data, and selective classification on a real-world medical dataset—where it consistently outperforms state-of-the-art credal classifiers (Table 1, Figures 3–5). Moreover, the ablation in Table 4 confirms that performance remains stable across a range of $\delta_G$ values, demonstrating that the method is not sensitive to this specific choice.
>
> **W2:** Thanks for raising this point. Our implementation follows established benchmarks, with details provided in Appendix B. *As stated and promised in our submission (Appendix B), the complete code will be released publicly upon publication.*
>
> **Q1:** Our ablation study (Sec. 4.2) shows that as $M$ increases ($M \in \lbrace 5, 10, 15 \rbrace$), *CreDRO* consistently improves and performs best across OOD detection benchmarks, as larger $M$ yield more informative credal sets for EU quantification. Computational cost scales linearly with $M$, with no overhead beyond training each member with the DRO objective.
>
> **Q2:** EN-DRO and DE compute EU via approximate mutual information (Eq. 5), whereas CreDRO derives EU from the credal set as the difference between upper and lower entropy (Eq. 10), which explicitly captures EU from a *set* of plausible distributions rather than relying on disagreement among finite predictive probability vectors. OOD detection is then performed as binary classification using this EU estimate as the prediction score, evaluated via AUROC (details in lines 246–249 and 255–261). We will make this more prominent in the revision.
>
> **Q3:** Table 1 results follow the 3-run protocol of [4] for fair comparison. For our additional experiments, we use more runs where feasible: 15 runs for corrupted data (Sec. 4.5) and 5 runs for the large-scale medical dataset (Sec. 4.6), where computational cost is significant.
>
> **Q4:** Table 4 confirms that CreDRO's performance is not sensitive to the choice of $\delta_G$​, with detailed analysis provided in Appendix D. Since performance remains consistently strong across all evaluated benchmarks, we recommend practitioners use the default value $\delta_G=0.5$, which eliminates the need for extensive hyperparameter tuning.
>
> **Q5.** While EN-DRO occasionally closes to (*is still worse than*) *CreDRO* in Table 4 for $M=20$, our broader evaluation, e.g., Figures 2-5, shows that *CreDRO* consistently and visibly outperforms EN-DRO across a wider range of settings. This is expected: the credal set explicitly models the *set* of plausible distributions, yielding a more informative epistemic uncertainty estimation than the disagreement among finite predictive distributions used by EN-DRO, and thus better separating ID from OOD samples.
>
> *We hope to have thoroughly addressed your concerns and trust that this provides stronger support for our paper.*
>
> [1] Daniel Levy et al. Large-Scale Methods for  Distributionally Robust Optimization. NeurIPS 2020.
>
> [2] Anurag Singh et al. Domain Generalisation via Imprecise Learning. ICML 2024.
>
> [3] Zeyi Huang et al. The Two Dimensions of Worst-Case Training and Their Integrated Effect for Out-of-Domain Generalization. CVPR 2022.
>
> [4] Timo Löhr et al. Credal Prediction based on Relative Likelihood. NeurIPS 2025.

---

> > ### Author Rebuttal · Reviewer_dHh4 · 2026-04-02
> >
> > Thank you very much for these detailed answers. All my concerned have been addressed. As a consequence, I raise my score.

---

> > > ### Author Response · Authors · 2026-04-03
> > >
> > > Dear Reviewer,
> > >
> > > We truly thank you for your positive feedback on our rebuttal. We are grateful for your support and acknowledgment.
> > >
> > > Thank you once again for your time, effort, and comments. We wish you all the best.
> > >
> > > Best regards,
> > >
> > > The Authors

---

### Decision · Program_Chairs · 2026-04-30

**Decision:**

Accept (spotlight)

**Comment:**

This paper presents CreDRO, a method for modeling epistemic uncertainty by moving beyond the standard view that uncertainty is primarily captured through disagreement induced by random initialization. Instead, the paper defines epistemic uncertainty through disagreement among models trained under varying degrees of relaxation of the i.i.d. assumption, implemented via distributionally robust optimization. This is a well-motivated and original perspective, and the resulting framework is both technically interesting and practically appealing because it works with standard neural architectures and produces credal predictions without requiring architectural modification. The empirical evaluation is also strong: the paper demonstrates consistent gains across a broad set of downstream tasks, including OOD detection, corruption robustness, and selective classification in a medical setting. Overall, the work makes a meaningful contribution to uncertainty quantification, with a clear methodological novelty and promising practical relevance.

The main concerns raised during review centered on the theoretical grounding of the DRO parameterization, the justification for top-loss selection as more than a heuristic, the potential confounding effect of label noise, and whether the box credal approximation might be overly conservative. In my view, the rebuttal addressed these points effectively. The authors clarified the connection between their objective and the CVaR-based DRO formulation, strengthening the theoretical basis of the method, and they added empirical evidence showing that the gains are not simply due to arbitrary sample reweighting. The additional analyses on interpolation strategies, label-noise robustness, and ID/OOD uncertainty behavior substantially improved confidence in the method and resolved most of the initial weaknesses identified by the reviewers. While some limitations remain, particularly regarding broader theoretical characterization and validation under even wider classes of shifts, they do not outweigh the paper’s strengths. Given the originality of the idea, the practical value of the framework, and the thorough rebuttal, I support acceptance.